# Bridged Clustering for Representation Learning: Semi-Supervised Sparse Bridging

## Abstract

We introduce **Bridged Clustering**, a semi-supervised framework to learn predictors from any *unpaired* input $\mathcal{X}$ and output $\mathcal{Y}$ dataset. Our method first clusters $\mathcal{X}$ and $\mathcal{Y}$ independently, then learns a sparse, interpretable bridge between clusters using only a few paired examples. At inference, a new input $x$ is assigned to its nearest input cluster, and the centroid of the linked output cluster is returned as the prediction $\hat{y}$. Unlike traditional SSL, Bridged Clustering *explicitly* leverages output-only data, and unlike dense transport-based methods, it maintains a sparse and interpretable alignment. Through theoretical analysis, we show that with bounded mis-clustering and mis-bridging rates, our algorithm becomes an effective and efficient predictor. Empirically, our method is competitive with SOTA methods while remaining simple, model-agnostic, and highly label-efficient in low-supervision settings.

## 1 Introduction

Analyzing modern datasets often involves reconciling rich but unaligned data collected for different purposes. For example, researchers may inherit a large morphological collection of museum specimen records alongside an independent genetic sequence repository such as GenBank (Sayers et al., 2019), without reliably coordinated specimen identifiers to join them (Mulcahy, 2022). This scenario is common across domains such as biomedicine, ecology, and the social sciences, where modern data ecosystems rarely provide perfectly paired records for every variable of interest (Putrama & Martinek, 2024; Wafa'Za'al Alma'aitah et al., 2024). Building supervised models in these settings is challenging, often requiring expensive relabeling or non-scalable heuristic alignment strategies.

*Semi-supervised learning* (SSL) is a natural approach when labeled pairs are scarce. By leveraging large sets of unlabeled inputs, SSL methods such as graph-based label propagation (Zhu & Ghahramani, 2002), consistency regularization techniques like Mean Teacher (Tarvainen & Valpola, 2017) and Temporal Ensembling (Laine & Aila, 2016), and hybrid approaches such as MixMatch (Berthelot et al., 2019) and FixMatch (Sohn et al., 2020) can dramatically reduce annotation costs. Similarly, transductive regressors like LapRLS (Belkin et al., 2006) have proven effective in settings with limited supervision. However, these approaches assume that each unlabeled example is an input $x$ whose label $y$ is merely hidden (Van Engelen & Hoos, 2020; Yang et al., 2022). This assumption breaks in the presence of output-only data, which is common in practice. Here "output-only" is defined relative to a user-specified prediction direction between two distinct modalities (e.g., $X \to Y$): it refers to samples for which only the $Y$-features are observed and no paired $X$ is available, while the complementary pool contains $X$-only samples. Output-only data includes, for example, unpaired DNA barcodes collected independently of morphological images (Ratnasingham et al., 2024), clinical outcome records separate from patient biosignals (Mark, 2016), satellite imagery without matching weather station logs, audio recordings lacking textual transcriptions, or microscopy images without corresponding gene-sequencing profiles. Therefore, to fully capitalize on available data, a predictive model must learn not only from unlabeled inputs but also from the structure and abundance of unlabeled outputs. Motivated by this challenge, we ask:

> *Can SSL be extended to incorporate rich, unlinked datasets of inputs and outputs for effective prediction, through simple clustering techniques?*

Several related frameworks have been proposed to connect disjoint input and output spaces. *Transport-based alignment* methods leverage output-only data via distributional coupling: entropic optimal transport with barycentric mapping (EOT-BM) uses a cross-domain cost $c(x, y)$ to couple $\mathcal{X}$ and $\mathcal{Y}$ and predicts by barycentric projection, while *Gromov–Wasserstein* (GW) aligns intra-space geometries when no cross-domain metric is available (Cuturi, 2013; Mémoli, 2011; Peyré et al., 2019). Unmatched regression and data fusion tackle separate samples of $X$ and $Y$ under structural assumptions (e.g., linear or monotone links, kernel embeddings) (Azadkia & Balabdaoui, 2024; Evans et al., 2021; Robbins et al., 2024; Hsu & Ramos, 2019; Chau et al., 2021). Generative methods such as mixture-of-regressions via EM (Dempster et al., 1977) can also be used to alternate imputations of missing inputs or outputs. Multi-view learning and co-training align modalities but require paired views per example (Blum & Mitchell, 1998; Bickel & Scheffer, 2004; Rey & Roth, 2012). Our approach differs by being *semi-supervised at the cluster level*: we independently structure each space using unpaired data, then resolve the input–output association with a small paired set, yielding a sparse, interpretable bridge and linear-time scaling in $n$ if $k \ll n$.

We introduce BRIDGED CLUSTERING, a simple and general SSL algorithm that builds a predictor from unpaired datasets using only a small set of matched examples (Algorithm 1). In a nutshell, it:

1. Independently cluster the input and output spaces.
2. Learn a *cluster-level bridge* from the small paired set.
3. Predict by assigning a new input to its nearest input cluster and returning the centroid of the linked output cluster.

BRIDGED CLUSTERING is *model-agnostic*, as any embedding model (e.g., ResNet, BERT) and standard clustering algorithm (e.g., $k$-means, spectral) can be used, and requires as few as one paired example per active input cluster. By leveraging structure in both modalities and bridging them through a minimal set of task-specific labeled pairs, BRIDGED CLUSTERING enables accurate prediction without the need for large-scale supervision. Moreover, since both unlabeled $\mathcal{X}$ and unlabeled $\mathcal{Y}$ contribute to refining the cluster centroids, the estimator's risk decreases with the total pool size, even in regimes with extremely limited pairing. We demonstrate the effectiveness of this method across diverse domains: vision, language, and bioinformatics.

**Design simplicity.** Our method is minimal, using standard clustering in both spaces and a sparse majority-vote bridge learned from a handful of labeled pairs. It is model-agnostic at both the encoder and clustering algorithm levels, and interpretable through cluster-to-cluster correspondences.

**Positioning.** Like transport-based alignments and other unmatched regression methods, our method leverages both input-only and output-only data; unlike them, we use a tiny paired set to learn a sparse, cluster-to-cluster bridge with explicit error controls. Our contributions are:

- **Robust learning under sparse supervision** by using large pools of unlabeled inputs and outputs to refine cluster centroids and training the bridge with only a few paired examples.
- **Leveraging independently collected heterogeneous data** by unifying disjoint input and output sources, so that structure in one modality informs prediction in the other.
- **Bidirectional inference across domains** via a cluster-level mapping that enables prediction of outputs from inputs *and* inputs from outputs without additional paired data.
- **Simplicity, interpretability, and efficiency.** The pipeline uses only standard clustering and a majority-vote bridge, yielding a sparse, interpretable cluster map and centroid predictors. This design is model-agnostic (plug-in encoders and clustering subroutines) and leads to linear-time scaling once $C \ll n$, as formalized in our runtime analysis (Section 5.3).

Our findings suggest a new research axis for SSL: *jointly exploiting input-only and output-only data by independent clustering*. BRIDGED CLUSTERING opens the door to leveraging the many unpaired available repositories without incurring prohibitive annotation costs.[1]

## 2 RELATED WORK

**Semi-supervised and Self-supervised Learning:** Early graph-based methods augment labeled data with unlabeled inputs by diffusing labels over a similarity graph on $\mathcal{X}$ (Zhou et al., 2003); Examples include Gaussian random-walk propagation and Laplacian Regularized Least Squares

---

[1]We release code and data: `https://anonymous.4open.science/r/bc-4B52/`.

(LapRLS; Belkin et al. (2006)). Modern consistency-regularization methods like Temporal Ensembling (Laine & Aila, 2016) and the Mean Teacher model (Tarvainen & Valpola, 2017) improve predictions by enforcing stability under input perturbations. MixMatch (Berthelot et al., 2019) unified label guessing with MixUp to encourage low-entropy regions, with FixMatch (Sohn et al., 2020) simplifying this via weak/strong augmentations with confidence-thresholded pseudo-labels. Dedicated SSL methods for regression remain rare. Twin Neural Network Regression (TNNR) (Wetzel et al., 2022) uses pairwise difference targets and unpaired anchor loops for self-consistency. Recent advances include Uncertainty-Consistent Variational Model Ensembling UCVME (Dai et al., 2023), which models noise for calibrated pseudo-labels, and RankUp (Huang et al., 2024), framing regression as an auxiliary ranking task to leverage classification SSL techniques. Graph neural networks (GNNs) provide additional transductive baselines. Graph Convolutional Networks (GCN; Kipf & Welling (2016)) combine neighborhood aggregation with partial supervision, while Adaptive Diffusion Convolution (ADC; Zhao et al. (2021)) learns hop-wise diffusion weights to mitigate over-smoothing. Unlike BRIDGED CLUSTERING, these methods only use unlabeled inputs and cannot leverage external output-only data.

Self-supervised contrastive methods like SimCLR (Chen et al., 2020) involve unlabeled multi-view augmentations, and deep clustering methods such as SCAN (Van Gansbeke et al., 2020) and NNM (Dang et al., 2021) also rely on multi-view consistency. These approaches use multiple augmented views, but operate within a single input modality, while our model jointly targets inputs and outputs.

**Multi-View and Dependency-Seeking Clustering:** Co-training trains two classifiers on disjoint feature views, exchanging pseudo-labels to leverage unlabeled data given conditional independence (Blum & Mitchell, 1998). Later works explore PAC and guarantees (Dasgupta et al., 2001; Balcan & Blum, 2010; Balcan et al., 2004). Dependency-Seeking Clustering (DSC) groups multi-view data by maximizing inter-view dependence (Rey & Roth, 2012). Bayesian mixtures of local CCAs (Viinikanoja et al., 2010) and Canonical Least-Squares similarly partition data via mixtures of linear sub-models (Lei et al., 2017). Unlike BRIDGED CLUSTERING, all require paired views and cannot use disjoint $\mathcal{X}$- or $\mathcal{Y}$-only corpora.

**Regression with Unmatched or Permuted Samples:** Recent statistical work on unmatched regression addresses separate, unordered datasets of $\mathcal{X}$ and $\mathcal{Y}$. Azadkia & Balabdaoui (2024) cast linear regression as a deconvolution problem, establishing consistency and parametric rates. Others handle monotone or isotonic structures (Carpentier & Schlüter, 2016; Rigollet & Weed, 2019). Bayesian data fusion approaches use kernel mean embeddings (Hsu & Ramos, 2019; Chau et al., 2021). Mixture-of-Regressions can be fit via Expectation-Maximization (Dempster et al., 1977), which in unmatched settings imputes missing $\mathcal{X}$ or $\mathcal{Y}$ jointly. Although some of these approaches unify unpaired and paired data, they do not leverage cluster geometry to sharpen predictions or produce explicit sparse cluster-level associations, setting BRIDGED CLUSTERING apart.

**Distributional alignment via Optimal Transport and GW.** Optimal transport (OT) aligns empirical measures across domains using a cross-space cost $c(x, y)$ and can produce predictions via *barycentric mapping* of the learned coupling; entropic regularization enables fast Sinkhorn solvers (Cuturi, 2013; Peyré et al., 2019). When no cross-domain metric is available, *Gromov–Wasserstein* (GW) matches intra-space distances to recover a soft correspondence between metric-measure spaces (Mémoli, 2011; Peyré et al., 2016). Both families exploit output-only data through distributional structure. Our approach differs in using a small paired set to resolve mode identities and learning an explicit, sparse cluster bridge that scales linearly once $k \ll n$, with guarantees tailored to clustering-based estimation rather than global transport objectives. Whereas OT/GW learn dense, global couplings, our bridge is *explicit and sparse*, trading global transport for cluster-level interpretability and linear-time inference once $C \ll n$.

**Cross-Domain Mapping without Parallel Data:** In NLP, unsupervised machine translation aligns monolingual embeddings and relies on back-translation to bridge languages (Lample et al., 2017). In computer vision, methods like CycleGAN learn bidirectional mappings between image domains via cycle-consistency constraints (Zhu et al., 2017; Huang et al., 2018). Though operating without paired instances, these systems train end-to-end neural generators and require reconstructive losses in both directions, while BRIDGED CLUSTERING is model-agnostic and learns only a sparse cluster-level association matrix using few labeled pairs.

---

**Algorithm 1** BRIDGED CLUSTERING

---

**Input**: Input-only set $\mathcal{X} = \{x_1, x_2, \ldots, x_{|\mathcal{X}|}\}$; Output-only set $\mathcal{Y} = \{y_1, y_2, \ldots, y_{|\mathcal{Y}|}\}$;
sparse supervised set $\mathcal{S} = \{(x_i', y_i')\}_{i=1}^k$; Test-set $\mathcal{X}_{\text{test}} = \{x_1, x_2, \ldots, x_{|\mathcal{X}_{\text{test}}|}\}$
(Note: $\mathcal{X}_{\text{test}} = \mathcal{X}$ in transductive setting)
**Output**: $\hat{\mathcal{Y}} = \{\hat{y}_1, \hat{y}_2, \ldots, \hat{y}_{|\mathcal{X}_{\text{test}}|}\}$
 1: (Optional) Enlarge $\mathcal{X}$ with the $x$-features of $\mathcal{S}$, enlarge $\mathcal{Y}$ with the $y$-features of $\mathcal{S}$.
 2: **Clustering in $\mathcal{X}$:**
 3: Apply a clustering algorithm to $\mathcal{X}$ to obtain cluster assignments $\mathcal{C}_\mathcal{X}$.
 4: **Clustering in $\mathcal{Y}$:**
 5: Apply a clustering algorithm to $\mathcal{Y}$ to obtain cluster assignments $\mathcal{C}_\mathcal{Y}$.
 6: **Bridge Learning:**
 7: Using the supervised set $\mathcal{S}$, learn a mapping $\hat{A}_{x \to y}$ between clusters in $\mathcal{X}$ and clusters in $\mathcal{Y}$.
 8: **Prediction:**
 9: **for** each sample $x_i$ in $\mathcal{X}_{\text{test}}$ **do**
10:     Assign $x_i$ to a cluster $c_x = \mathcal{C}_{\mathcal{X}_{\text{test}}}(x_i)$.
11:     Find corresponding cluster in $\mathcal{Y}$: $c_y = \hat{A}_{x \to y}(c_x)$.
12:     Predict $\hat{y}_i$ as the centroid of cluster $c_y$ in $\mathcal{Y}$.
13: **end for**
14: **return** Predicted outputs $\hat{\mathcal{Y}}$ for test inputs

---

## 3 PROBLEM FORMULATION

We formalize BRIDGED CLUSTERING's prediction task and identify conditions under which it achieves bounded excess risk, even with limited paired data. Without loss of generality, we focus on predicting an output $Y$ from a given input $X$; the reverse direction (predicting $X$ from $Y$) is symmetric and follows by swapping roles. We now introduce key assumptions and notations.

Consider a population of samples partitioned into $C$ latent groups. While the criteria defining these groups are unknown, we assume the existence of $C$ distinct categories. Let $T \in [C] := \{1, \ldots, C\}$ be a latent categorical variable drawn with probabilities $(\pi_1, \ldots, \pi_C)$. For each group $t$, let $P_t^X$ denote the distribution over input features and $P_t^Y$ the distribution over output features.

Conditional on $T = t$ we observe a pair: $(X, Y) \mid T = t \sim P_t^X \times P_t^Y$, and let

$$\mu_t^X := \mathbb{E}[X \mid T = t], \ \mu_t^Y := \mathbb{E}[Y \mid T = t].$$

Our method takes as input the following training data:

- An *input pool* $\mathcal{X} = \{x_i\}_{i=1}^{n_X} \subseteq \mathbb{R}^d$ of $n_X$ i.i.d. draws from the marginal $P^X = \sum_t \pi_t P_t^X$;

- an *output pool* $\mathcal{Y} = \{y_i\}_{i=1}^{n_Y} \subseteq \mathbb{R}^{d'}$ of $n_Y$ i.i.d. draws from the marginal $P^Y = \sum_t \pi_t P_t^Y$;

- a *small paired sample* $\mathcal{S} = \{(x_j', y_j')\}_{j=1}^k$, with $k \ll n_X, n_Y$, drawn i.i.d. from joint $P = \sum_t \pi_t(P_t^X \times P_t^Y)$.

## 4 BRIDGED CLUSTERING ALGORITHM

BRIDGED CLUSTERING (Algorithm 1), first independently clusters the input space $\mathcal{X}$ and the output space $\mathcal{Y}$. Next, it learns a bridge mapping $\hat{A}$ from the paired samples, where each input cluster is matched to an output cluster via majority vote. Finally, for inference, a new input $x$ is assigned to its nearest input cluster, and the prediction $\hat{y}$ is given by the centroid of the linked output cluster.

**Cluster assignments:** We first run a $k$-clustering algorithm to partition the dataset into $k$ groups separately on $\mathcal{X}$ and $\mathcal{Y}$ using $C$ centers: $\hat{c}_X : \mathbb{R}^d \to [C], \hat{c}_Y : \mathbb{R}^{d'} \to [C]$. Because cluster labels are only defined up to a permutation of indices, we evaluate their quality via the permutation-invariant mis-clustering rates:

$$\varepsilon_X := \min_{\sigma:[C] \to [C]} \Pr[\sigma(\hat{c}_X(X)) \neq T], \varepsilon_Y := \min_{\sigma:[C] \to [C]} \Pr[\sigma(\hat{c}_Y(Y)) \neq T].$$

Let $\sigma_X, \sigma_Y$ denote permutations that attain these minima.

**Bridging from few pairs:** On the $k$ paired points, we *match* input-space clusters to output-space clusters by majority vote. For every input cluster $a$, mapping $\hat{A}(a)$ := $\arg\max_{b \in [C]} \sum_{(x', y') \in \mathcal{S}} \mathbf{1}\{\hat{c}_X(x') = a, \ \hat{c}_Y(y') = b\}$. For a random latent label $T$, we call $\varepsilon_B := \Pr\left[\hat{A}(T) \neq T\right]$ the *mis-bridging rate*. Algorithm 1 predicts a fresh output by returning the empirical centroid of the output cluster indicated by $\hat{A} \circ \hat{c}_X$. Let $\widehat{\mu}_b^Y$ be the empirical mean of $\mathcal{Y}$-points assigned to cluster $b$, then $\hat{Y}(x) := \widehat{\mu}_{\hat{A}(\hat{c}_X(x))}^Y$.

# 5  ALGORITHMIC ANALYSIS

This section provides a theoretical understanding of BRIDGED CLUSTERING, identifying key factors governing its algorithmic effectiveness and efficiency.

## 5.1  INFERENCE EFFECTIVENESS

For a random sample $(X, Y) \mid T$, the BRIDGED CLUSTERING Algorithm is tasked with retrieving an estimation of $Y$ based on $X$. To analyze the effectiveness of inference, we atomize the process:

$$X \xrightarrow[\varepsilon_X]{\hat{c}_X, (\sigma_X)} \hat{T} \xrightarrow[\varepsilon_B]{\hat{A}} \hat{\mu}_{\hat{T}}^Y = \hat{Y} \xrightarrow[\varepsilon_Y, D_Y]{\hat{c}_Y, (\sigma_Y)} Y$$

Our model first implicitly predicts a latent label $\hat{T}$ for the sample input $X$ by using $\hat{c}_X$ to assign it to the closest input cluster. Then, using the learned cluster-to-cluster mapping $\hat{A}$, our model reaches the corresponding output cluster centroid $\hat{\mu}_{\hat{T}}^Y$. From there, the distance of our estimated output $\hat{Y}$ from the actual output $Y$ is mainly determined by two factors: whether $Y$ is assigned by $\hat{c}_Y$ to the correct output cluster $T$ (bounded by $\varepsilon_Y$), and—if the assignment is correct—the in-cluster distance between the $Y$ and its cluster centroid; we define this distance as $D_Y$.

Throughout the process, $\hat{A}, \hat{c}_X, \hat{c}_Y$ are the key functions solidified during model training, learned from the supervised data, unsupervised inputs, and unsupervised outputs, respectively. The success of each step during inference is quantified through $\varepsilon_X, \varepsilon_Y, \varepsilon_B, D_Y$. Correspondingly, as our model becomes robust in input-assignment, output assignment, and bridging accuracy, the $\varepsilon_X, \varepsilon_Y, \varepsilon_B$ are actively controlled so that the model makes more accurate inferences.

## 5.2  RISK CONTROL

We quantify how representation quality and scarce supervision govern the three error sources in BRIDGED CLUSTERING: input mis-clustering ($\varepsilon_X$), output mis-clustering ($\varepsilon_Y$), and mis-bridging ($\varepsilon_B$). The high-level takeaways are that (1) stronger embeddings that enlarge inter-clustering separation directly reduce $\varepsilon_X$ and $\varepsilon_Y$, and (2) once clusters are reliable, even a small handful of paired examples suffice to learn the sparse cluster bridge with exponentially small error.

**Mis-clustering under sub-Gaussian mixtures.** Suppose the input and output distributions are $\sigma^2$–sub-Gaussian with means $\mu_t^X, \mu_t^Y$. Define the minimum separation $\Delta_X = \min_{t \neq s \in [C]} \|\mu_t^X - \mu_s^X\|$ and $\Delta_Y = \min_{t \neq s \in [C]} \|\mu_t^Y - \mu_s^Y\|$. Classical results on Lloyd's algorithm show that, under good initialization, $C$-means clustering on $n_X$ unlabeled inputs satisfies

$$\varepsilon_X \leq \exp\left(-\frac{\Delta_X^2}{16\sigma^2}\right) \quad \text{for all } s \geq 4 \log n_X$$

with probability at least $1 - \nu - \frac{4}{n_X} - 2\exp(-\Delta_X/\sigma)$, where $\nu$ is the initializer failure rate (Lu & Zhou, 2016). An identical bound holds for $\varepsilon_Y$.

Thus, any improvement in embeddings that increases $\Delta_X$ or $\Delta_Y$ sharpens the bound without changing the algorithm. Consistent with this theory, we observe a strong negative correlation between cluster quality (AMI) and predictive loss (Appendix K).

**Efficient bridging from few pairs.** Once the mis-clustering rates $\varepsilon_X, \varepsilon_Y$ are controlled, learning the bridge becomes simple. For a supervised pair $(X,Y)|T \in \mathcal{S}$, consider the events $\{\sigma_X(\hat{c}_X(X)) = T\}$ and $\{\sigma_Y(\hat{c}_Y(Y)) = T\}$, which correspond to correct cluster assignments in the input and output spaces, controlled via $\varepsilon_X, \varepsilon_Y$. If both events hold with sufficiently high probability ($> 50\%$), then a majority vote over just a few labeled pairs will suffice to recover the correct cluster-to-cluster assignment, resulting in a low mis-bridging error $\varepsilon_B$. Empirically, we find that even one or two examples per cluster are enough for reliable bridging, explaining why BRIDGED CLUSTERING remains effective with less than $1\%$ paired supevision (Sec. 6).on.

### 5.3 RUNTIME AND MEMORY COMPLEXITY

We break down the cost of BRIDGED CLUSTERING into clustering, bridge learning, and inference. To remain solver-agnostic, let $f(n, C, d)$ denote the time our chosen algorithm takes to cluster $n$ points in $\mathbb{R}^d$ into $C$ clusters. For BRIDGED CLUSTERING:

$$\text{training} = f(n_X, C, d) + f(n_Y, C, d') + O(k), \qquad \text{per-query inference} = O(Cd + d'),$$

with memory $O(n_X d + n_Y d' + C(d+d') + C^2)$.[2] Instantiating $f$: Lloyd/$k$-means++ gives $O(nCdI)$ where $I$ is the number of iterations; mini-batch $k$-means $O(mCdI)$ with batch $m \ll n$; pruned $k$-means $O(nC'dI)$ with $C' \leq C$ effective distance calls. Thus, training is linear in $n_X, n_Y$ when $C \ll n$, and inference requires only $O(C)$ distance computations instead of dense cross-space objects.

Now, as we found in our experiments in Section 6, the transport-based methods, EOT and GW, are the closest competitors to our method. We compare our method asymptotically to the standard implementations of EOT and GW. This analysis is conducted in the transductive setting, though the asymptotic conclusions extend similarly to the inductive case. Let $S$ denote the number of Sinkhorn iterations in EOT, and $T_{\text{GW}}$ the number of outer GW iterations; we report only the dominant terms.

| Method | Training time | Memory |
|---|---|---|
| BC (ours) | $f(n_X, C, d) + f(n_Y, C, d') + O(k)$ | $O(n_X d + n_Y d' + C(d+d') + C^2)$ |
| EOT | $O(S\,n_X n_Y)$ | $O(n_X n_Y)$ |
| GW | $O(T_{\text{GW}}(n_X^2 n_Y + n_X n_Y^2))$ | $O(n_X^2 + n_Y^2 + n_X n_Y)$ |

Since both transport-based methods involve Sinkhorn iterations and dense couplings, they incur at least quadratic cost in both time and memory, which scales quickly with data increase. In contrast, BRIDGED CLUSTERING with a fast clustering routine (e.g., mini-batch $k$-means) achieves training time linear in $n_X, n_Y$ and memory linear in the raw data size plus a (sparse) $C \times C$ bridge, making it more efficient in both runtime and memory than transport-based baselines in large-pool regimes.

## 6 EXPERIMENTS

We now turn to empirically test the predictive accuracy of BRIDGED CLUSTERING. We use fully paired datasets with known input-output correspondences ($(x_i, y_i) \in \mathcal{X} \times \mathcal{Y}$). These datasets allow us to simulate the disjoint setting by hiding most of the pairings and exposing only a small, randomly selected subset during training. This controlled setup enables us to measure performance as a function of the number of labeled pairs, while retaining access to ground-truth labels for evaluation.

### 6.1 DATA

We apply BRIDGED CLUSTERING on four multimodal datasets that connect distinct feature spaces: the *BIOSCAN-5M* dataset that pairs specimen images with DNA barcode sequences of insect samples (Gharaee et al., 2024), the *WIT* dataset with Wikipedia Image-Text pairings (Srinivasan et al., 2021), as well as the *Flickr30k* dataset (Young et al., 2014) and the *COCO* dataset (Lin et al., 2014) that connect everyday images with human-generated descriptions. We randomly sample a number of data groups to include in each run of our experiments, e.g., selecting 5 insect species from Bioscan, or 7 object categories from COCO, etc. We detail the data, processing, and mapping in Appendix G.

---

[2]If only a mapping $[C] \to [C]$ is stored, the bridge term is $O(C)$; $O(C^2)$ is a safe upper bound for storing a sparse $C \times C$ vote/mapping matrix.

## 6.2 EXPERIMENTAL OVERVIEW

We evaluate BRIDGED CLUSTERING's ability to predict the output features $\mathcal{Y}$ (e.g., DNA sequences or text captions) from the input features $\mathcal{X}$ (e.g., images). For each dataset, we simulate a disjoint data setting by designating a small subset of datapoints as the supervised set, which contains paired input-output examples. The remaining datapoints are divided into two unpaired sets: one with input-only features and the other with output-only features, mimicking independent data sources. During training, the model leverages 3 sources of information: (1) unpaired input features ($\mathcal{X}$), (2) unpaired output features ($\mathcal{Y}$), and (3) a small set of paired examples $(x_i, y_i) \in \mathcal{X} \times \mathcal{Y}$. For evaluation, we assess the model's ability to recover the missing output features $\mathcal{Y}$ of the input-only datapoints $\mathcal{X}$.

One of the advantages of BRIDGED CLUSTERING is its bidirectionality. Thus, in addition to predicting $\mathcal{Y}$ from $\mathcal{X}$, we also test the inverse by predicting input features $\mathcal{X}$ from output features $\mathcal{Y}$.

We compute the Mean Squared Error (MSE) between the predicted and the ground-truth output features. In image captioning tasks, outputs are text embeddings; thus, we map each predicted embedding to its nearest neighbor in the unpaired output set. We then compute the MSE between this retrieved embedding and the ground-truth caption embedding (see details in Appendix G).

## 6.3 BASELINES

We compare BRIDGED CLUSTERING against a comprehensive suite of baselines spanning supervised learning, classical transductive methods, graph-based models, consistency-regularized deep learners, pairwise regression approaches, and unmatched regression techniques. Together, they span the full landscape of SSL and related paradigms. All except KNN leverage unlabeled inputs and paired data. BRIDGED CLUSTERING, KMM, EM, EOT, GW incorporate unlabeled outputs as well.

- Supervised: K-nearest neighbors (KNN).
- Transductive SSL: Laplacian Regularized Least Squares (LapRLS) and Transductive SVR (TSVR; the standard Support-Vector Regression was adapted to semi-supervised setting).
- Graph-structured: Graph Convolutional Network (GCN).
- Consistency-regularized: FixMatch.
- Low-supervision regressors: Twin Neural Network Regression (TNNR), and Uncertainty-Consistent Variational Model Ensembling (UCVME).
- Unmatched-regression: Kernel Mean Matching (KMM), Mixture-of-Regressions via EM.
- Transport-based: Entropic OT with Barycentric Mapping (EOT), Gromov-Wasserstein (GW).

## 6.4 RESULTS

We evaluate BRIDGED CLUSTERING on four datasets, testing its ability to map $x \in \mathcal{X}$ to $y \in \mathcal{Y}$ and vice versa. Since BRIDGED CLUSTERING assumes data can be grouped discretely, each experiment samples a subset of groups (e.g., species in BIOSCAN). We test the robustness of BRIDGED CLUSTERING by sampling different numbers of data groups $\in \{3, 4, 5, 6, 7\}$ and varying the numbers of supervised samples per cluster $\in \{1, 2, 3, 4\}$[3], yielding $4 \times 5 = 20$ different settings per experiment.

Each setting is run 30 times with different seeds. Figure 1 shows MSE distributions, while Table 6.4 reports win-rates (lowest MSE across 600 runs per dataset). Additional results in Figures A.19-A.22. Overall, BRIDGED CLUSTERING outperforms all baselines across all datasets except WIT. We note the robustness of the results across transductive and inductive setups.

## 7 DISCUSSION

BRIDGED CLUSTERING outperforms baseline performance in three out of four datasets (we will discuss WIT's performance shortly). In these datasets, Wilcoxon signed-rank tests confirmed these findings are statistically significant (largest p-value $= 6\mathrm{e}{-4}$).

---

[3]Corresponds to $\{0.5\%, 1\%, 1.5\%, 2\%\}$ supervision for Bioscan and COCO (cluster size=200), and $\{4\%, 8\%, 12\%, 16\%\}$ for WIT and Flickr30k (cluster size=25).

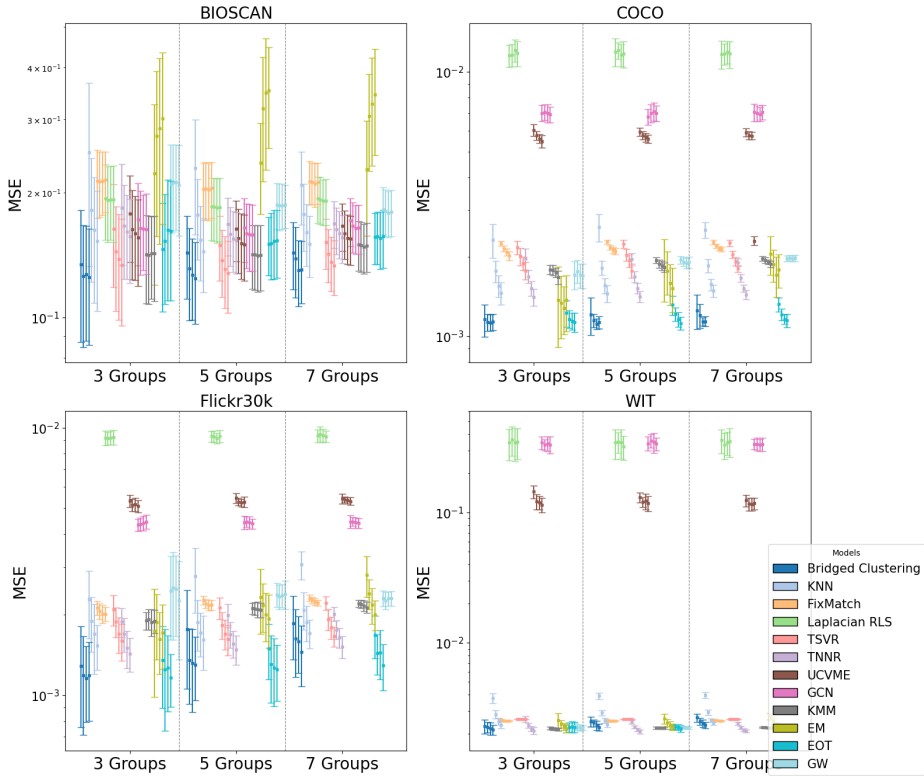

Figure 1: MSE distribution of different models in the inductive setting. The four distribution plots of the same color represent the settings with 1, 2, 3, and 4 supervised samples per cluster.

**Robust success:** Across both directions ($\mathcal{X} \to \mathcal{Y}$, $\mathcal{Y} \to \mathcal{X}$) in Bioscan, Flickr30k, and COCO, BRIDGED CLUSTERING achieves consistently low MSE, outperforming all baselines over varied cluster sizes and supervision levels. On WIT, it falls short of the best model but remains highly competitive, notably surpassing all SSL baselines by leveraging output-only data $\mathcal{Y}$. On average, BRIDGED CLUSTERING also outperforms KMM, EM, EOT, and GW, despite these methods using full training data, underscoring the strength of our simpler approach. While transport-based models are strong baselines, BRIDGED CLUSTERING offers greater interpretability and efficiency.

**Cluster quality and accuracy:** BRIDGED CLUSTERING's performance depends on recovering latent structure via independent clustering (cluster quality) and aligning clusters with limited supervision (bridging accuracy). Theoretically, inference quality is bounded by mis-clustering rates $\varepsilon_X, \varepsilon_Y$ and bridging accuracy $\varepsilon_B$, which together limit model risk. Empirically, cluster quality (adjusted mutual information) negatively correlates with model loss (Figure A.25). WIT's low mutual information suggests that weaker clustering explains its lower performance (Appendix K).

**Low supervision regime:** BRIDGED CLUSTERING performs well in low-supervision settings, as it requires as few as one labeled example per cluster to associate input and output clusters accurately. While increasing the number of supervised points can improve alignment accuracy, it has a limited effect on prediction quality. In other words, since the model always returns the output cluster centroid, further supervision does not refine individual predictions. In contrast, other models (excluding EOT) may benefit more from additional supervision.

**Higher risk with more data groups:** BRIDGED CLUSTERING's relative weakness in high-cluster regimes stems from its reliance on bijection input-output mapping. As cluster count increases, the space of possible mappings grows exponentially, making accurate alignment harder (See Figure 28 in Appendix K). Baseline methods without such explicit mappings prove more robust to increases in cluster count, highlighting BRIDGED CLUSTERING's tradeoff: it excels in low-supervision, low-cluster settings, but becomes more sensitive as structural complexity increases.

**(a) Transductive experiments**

| Exp. | BC | KNN | FM | LapR. | TSVR | TNNR | UCVME | GCN | KMM | EM | EOT | GW |
|---|---|---|---|---|---|---|---|---|---|---|---|---|
| BIOSCAN | **0.67** | 0.00 | 0.00 | 0.00 | 0.19 | 0.00 | 0.00 | 0.00 | 0.13 | 0.01 | 0.01 | 0.00 |
| BIOSCAN (rev.) | **0.63** | 0.00 | 0.00 | 0.00 | 0.00 | 0.00 | 0.00 | 0.00 | 0.01 | 0.01 | 0.36 | 0.00 |
| COCO | **0.71** | 0.00 | 0.00 | 0.00 | 0.00 | 0.00 | 0.00 | 0.00 | 0.00 | 0.03 | 0.26 | 0.00 |
| COCO (rev.) | **0.89** | 0.00 | 0.00 | 0.00 | 0.00 | 0.00 | 0.00 | 0.00 | 0.00 | 0.04 | 0.07 | 0.00 |
| Flickr30k | **0.56** | 0.00 | 0.00 | 0.00 | 0.00 | 0.04 | 0.00 | 0.00 | 0.01 | 0.02 | 0.36 | 0.01 |
| Flickr30k (rev.) | **0.71** | 0.00 | 0.00 | 0.00 | 0.13 | 0.00 | 0.00 | 0.00 | 0.03 | 0.00 | 0.10 | 0.04 |
| WIT | 0.12 | 0.00 | 0.00 | 0.00 | 0.00 | **0.32** | 0.00 | 0.00 | 0.28 | 0.03 | 0.16 | 0.10 |
| WIT (rev.) | 0.09 | 0.00 | 0.00 | 0.00 | 0.04 | **0.30** | 0.00 | 0.00 | 0.27 | 0.00 | 0.05 | 0.25 |

**(b) Inductive experiments**

| Exp. | BC | KNN | FM | LapR. | TSVR | TNNR | UCVME | GCN | KMM | EM | EOT | GW |
|---|---|---|---|---|---|---|---|---|---|---|---|---|
| BIOSCAN | **0.67** | 0.00 | 0.00 | 0.00 | 0.18 | 0.00 | 0.00 | 0.00 | 0.11 | 0.01 | 0.03 | 0.00 |
| BIOSCAN (rev.) | **0.61** | 0.00 | 0.00 | 0.00 | 0.00 | 0.00 | 0.00 | 0.00 | 0.01 | 0.01 | 0.37 | 0.00 |
| COCO | **0.70** | 0.00 | 0.00 | 0.00 | 0.00 | 0.00 | 0.00 | 0.00 | 0.00 | 0.03 | 0.28 | 0.00 |
| COCO (rev.) | **0.84** | 0.00 | 0.00 | 0.00 | 0.00 | 0.00 | 0.00 | 0.00 | 0.00 | 0.05 | 0.11 | 0.00 |
| Flickr30k | 0.45 | 0.01 | 0.00 | 0.00 | 0.00 | 0.04 | 0.00 | 0.00 | 0.01 | 0.03 | **0.46** | 0.01 |
| Flickr30k (rev.) | **0.60** | 0.01 | 0.00 | 0.00 | 0.00 | 0.01 | 0.00 | 0.00 | 0.03 | 0.01 | 0.32 | 0.03 |
| WIT | 0.12 | 0.00 | 0.00 | 0.00 | 0.00 | **0.32** | 0.00 | 0.00 | 0.28 | 0.03 | 0.16 | 0.10 |
| WIT (rev.) | 0.09 | 0.00 | 0.00 | 0.00 | 0.04 | **0.30** | 0.00 | 0.00 | 0.27 | 0.00 | 0.05 | 0.25 |

Table 1: Win-rates across transductive (a) and inductive (b) experiments. Bold indicates the best model per dataset. Win-rates are defined as the model with the lowest MSE across 600 runs in every dataset (30 randomized trials per setting). For example, if BRIDGED CLUSTERING achieves the lowest MSE among all models in 300 out of the 600 trials for some setting, its score is 0.50.

**Data-efficient learning:** BRIDGED CLUSTERING demonstrates that modular architectures leveraging unsupervised structure and lightweight alignment can match or exceed fully supervised models. By separating representation learning from supervision, it uncovers latent structure in input and output spaces and aligns them with minimal supervision. This approach is especially effective with limited labeled data, high domain variability, or distribution shifts, where monolithic models often overfit. Our results indicate that concept-level alignment between independently clustered spaces provides a strong inductive bias, enabling generalization without dense supervision.

**Conceptual appeal:** By decoupling structure discovery (unsupervised clustering) from cross-space association (a small, sparse bridge), BRIDGED CLUSTERING offers a simple blueprint for learning with abundant unpaired observations. This model-agnostic algorithm can be extended (hierarchical clusters, soft bridges, learned metrics) while preserving interpretability and efficiency.

## 8 ALGORITHMIC VARIANTS AND EXTENDED APPLICABILITY

We introduce BRIDGED CLUSTERING as one of the first and simplest blueprints for the line of research it motivates: the joint exploration of unsupervised input and output spaces.

To preserve conceptual simplicity, our main algorithmic design adopts the most general primitives: any clustering method for unsupervised learning and any mapping mechanism for sparse bridging. Our main experiments are designed to test BRIDGED CLUSTERING in data settings where clustering are known as effective, thereby isolating the effect of any inefficiency from unsupervised learning itself. As future work rely on more powerful unsupervised methods, similar algorithms can be designed on BRIDGED CLUSTERING's conceptual basis to tackle increasingly complex data settings.

To motivate how BRIDGED CLUSTERING could be adapted for more complex settings, we propose two algorithmic variants as a start, each targeting a range of data-centric challenges.

**Soft Bridging for Flexible Mapping**

When applying our algorithm to datasets without clearly-separated categories, clustering primitives may become prone to overlapping or miscounted clusters. Rigid one-to-one mapping, as in our previous formulation, becomes vulnerable to misattributed cluster under this setting. Hence, we design

a soft-bridging as our new mapping primitive: when using the small supervised set to build input-output cluster bridges, we build probabilistic bridges between every possible input-output cluster pair, dynamically weighing this edge based on the number of supervised points connecting them. Formally, let $w_{i \to j}$ denote the percentage of supervised points whose inputs fall in input cluster $i$ and whose labels lie in output cluster $j$, with $\sum_j w_{i \to j} = 1$. For an input $x$ assigned to cluster $i$, soft bridging predicts a weighted combination of all output cluster centroids: $\hat{y}(x) = \sum_{j=1}^{C} w_{i \to j} \bar{y}_j$.

This method reduces the harm of incorrect rigid bridges, especially relevant when clusters are not retrieved correctly: clustering algorithm may capture too few or too many clusters, or clusters may overlap each other, resulting in incorrect cluster-correspondence. Soft bridging doesn't hinge on exact cluster-correspondence, and is more robust to this challenge in cluster quality.

**Cluster-wise Regression For Refined Prediction**

Once an input is assigned to an input cluster $i$ and bridged to an output cluster $j$, our basic algorithm returns the output centroid $\bar{y}_j$, collapsing intra-cluster variation. To relax this rigidity, we propose cluster-wise regression, deployed as a third step after clustering and bridging. For each cluster pair, we use in-cluster supervised points to fit a linear map $f_{i \to j}(x)$ and predict

$$\hat{y}(x) = (1 - \alpha) \bar{y}_j + \alpha f_{i \to j}(x),$$

with $\alpha \in [0, 1]$ controlling the strength of the regression refinement. We instantiate two variants:

- **SUPERVISED REFINEMENT**: linear regression on the supervised pairs in $(i, j)$;
- **SUPERVISED+CENTROID REFINEMENT**: linear regression on the supervised pairs plus an artificial "supervised" point $(x_{\text{cen}}, y_{\text{cen}})$ given by our input/output cluster centroids.

Cluster-wise regression is especially valuable as template for extending BRIDGED CLUSTERING beyond very-low-supervision and conditionally independent settings. In regimes where the output $Y$ is conditionally linearly on the input $X$ given the clustered latent $Z$, this offers a direct way to refine our cluster-based predictor, if provided sufficient labeled pairs to fit these cluster-wise regressors.

**Experiments and Results with Challenging Data Conditions**

Targeting overlapping clusters, we designed a new experiment by sampling multi-labeled COCO classes that share at least one overlapping label with another sampled class, in contrast to our earlier use of single-labeled categories. For miscounted clusters, we designed another experiment in COCO by intentionally mis-specifying the number of clusters in our K-clustering primitives.

In the two experiments, both Soft-Bridging and Cluster-wise Regression (Supervised+Centroid Refinement) outperform our basic formulation under these challenging data settings, suggesting the adaptivity of BRIDGED CLUSTERING as an effective template for diverse algorithms that jointly exploit unsupervised inputs and outputs. See Figure 2 and 3 in Appendix A.

## 9 CONCLUSION & FUTURE WORK

BRIDGED CLUSTERING offers a new and simple vantage for SSL by elevating *output-only* data from a peripheral role to a central asset. By decoupling input from output representation and stitching them together with a sparse, interpretable bridge, our framework addresses a gap between classic input-centric SSL and the realities of modern, richly available but poorly aligned data ecosystems.

Our analysis shows that BRIDGED CLUSTERING can become a powerful predictor with bounded mis-clustering and mis-bridging rates. Empirically, it performs competitively among strong baselines across diverse domains and data modalities. The method is model-agnostic, computationally efficient, and supports bidirectional inference without retraining. These properties suggest a broader research direction: learning from abundant unpaired observations by first modeling each view independently, and then bridging them with minimal supervision.

Future work can explore domains with overlapping or hierarchical categories, extending to inductive settings, incorporating soft or probabilistic alignments, and integrating with unsupervised techniques such as manifold learning or self-supervised embeddings. These lead to more flexible SSL systems that can effectively harness the rich but misaligned data landscapes common in modern applications.

## 10 ETHICS STATEMENT

BRIDGED CLUSTERING primarily advances methodology in machine learning and does not directly involve human subjects or sensitive personal data. Nonetheless, we acknowledge that methods for automated clustering and prediction can be applied in real-world settings with potential social impact.

The design of BRIDGED CLUSTERING emphasizes interpretability and low supervision, which can help mitigate risks associated with opaque or biased models. By explicitly linking clusters and maintaining cluster-level transparency, users can inspect and understand the learned correspondences, reducing the likelihood of unintended harm from unexamined correlations.

We caution that, like any predictive system, applying BRIDGED CLUSTERING to high-stakes domains (e.g., hiring, law enforcement, healthcare) requires careful consideration of fairness, privacy, and potential biases in the data. Our experiments are conducted on publicly available benchmark datasets, and no sensitive or personally identifiable information is used.

## 11 REPRODUCIBILITY STATEMENT

We have taken several steps to ensure reproducibility. All datasets used in our experiments are publicly available, and preprocessing scripts are included in our supplementary materials. The implementation of BRIDGED CLUSTERING, including clustering, alignment, and evaluation routines, is fully documented and released alongside the paper.

Hyperparameters, random seeds, and experimental configurations for all reported results are explicitly listed in the code and appendix. Each experiment is repeated across multiple random seeds to quantify variability, and results are reported with appropriate statistics.

Additionally, we provide detailed instructions for reproducing both the main figures and tables, including cluster visualizations, MSE distributions, and win-rate calculations. This enables other researchers to verify our findings, test alternative design choices, and extend BRIDGED CLUSTERING to new datasets or tasks with minimal ambiguity.

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

# A  EXPERIMENT RESULTS FOR ALGORITHMIC VARIANTS

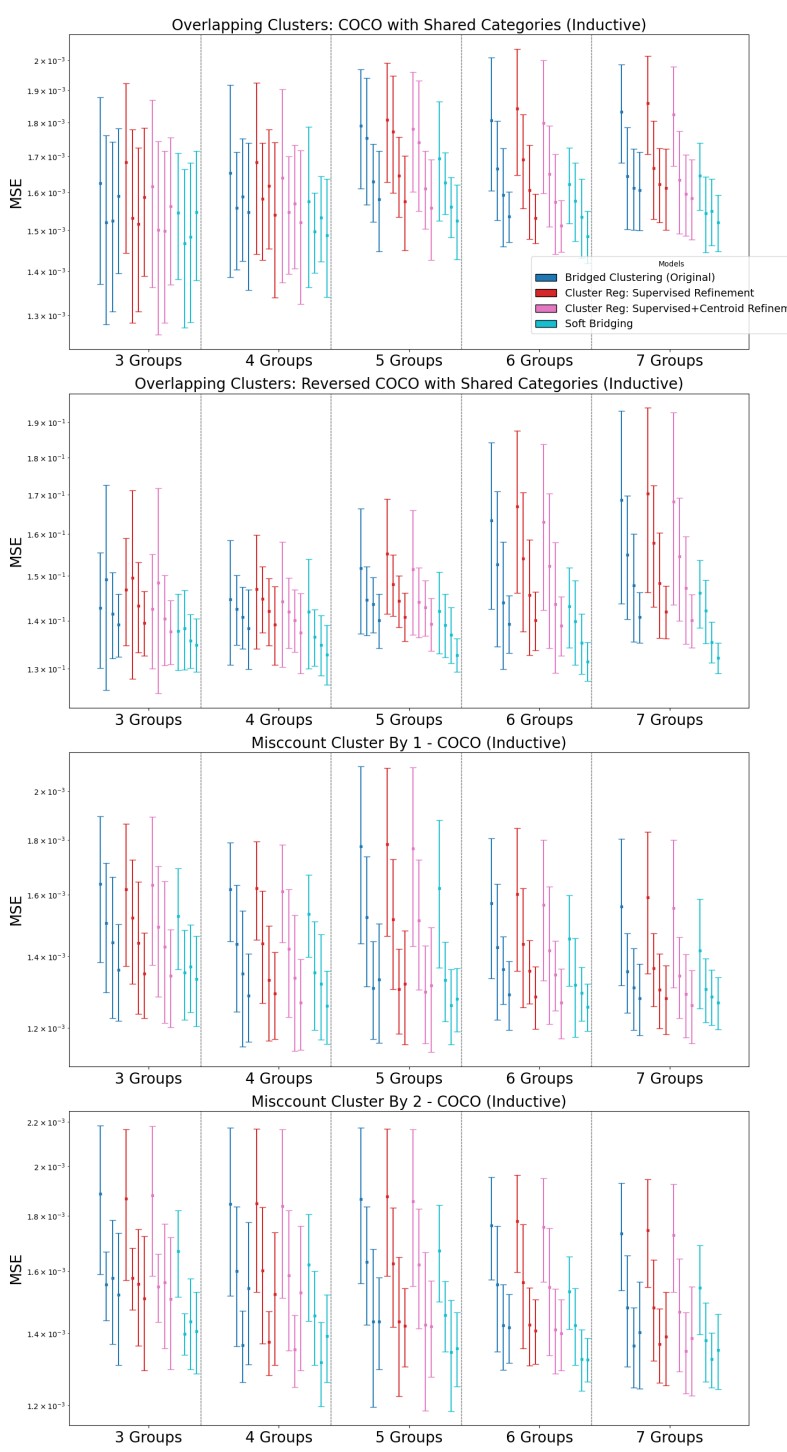

Figure 2: Inductive Experiments using Algorithmic Variants: The 4 distribution plots of the same color represent the settings with 1,2,3,4 supervised samples per cluster. Notice that Soft-Bridging is the overall strongest method in addressing both overlapping and misccounted clusters. For Cluster-wise Regression, Supervised Refinement alone performs unstably, but adding centroid to regressor fitting improves model performance, slightly outperforming the original Bridged Clustering model. As we set $\alpha = 0.25$, winning margin over is expectly small.

Here, miscounting refers to overcounting the number of clusters, and we note that undercounting experiments show similar results.

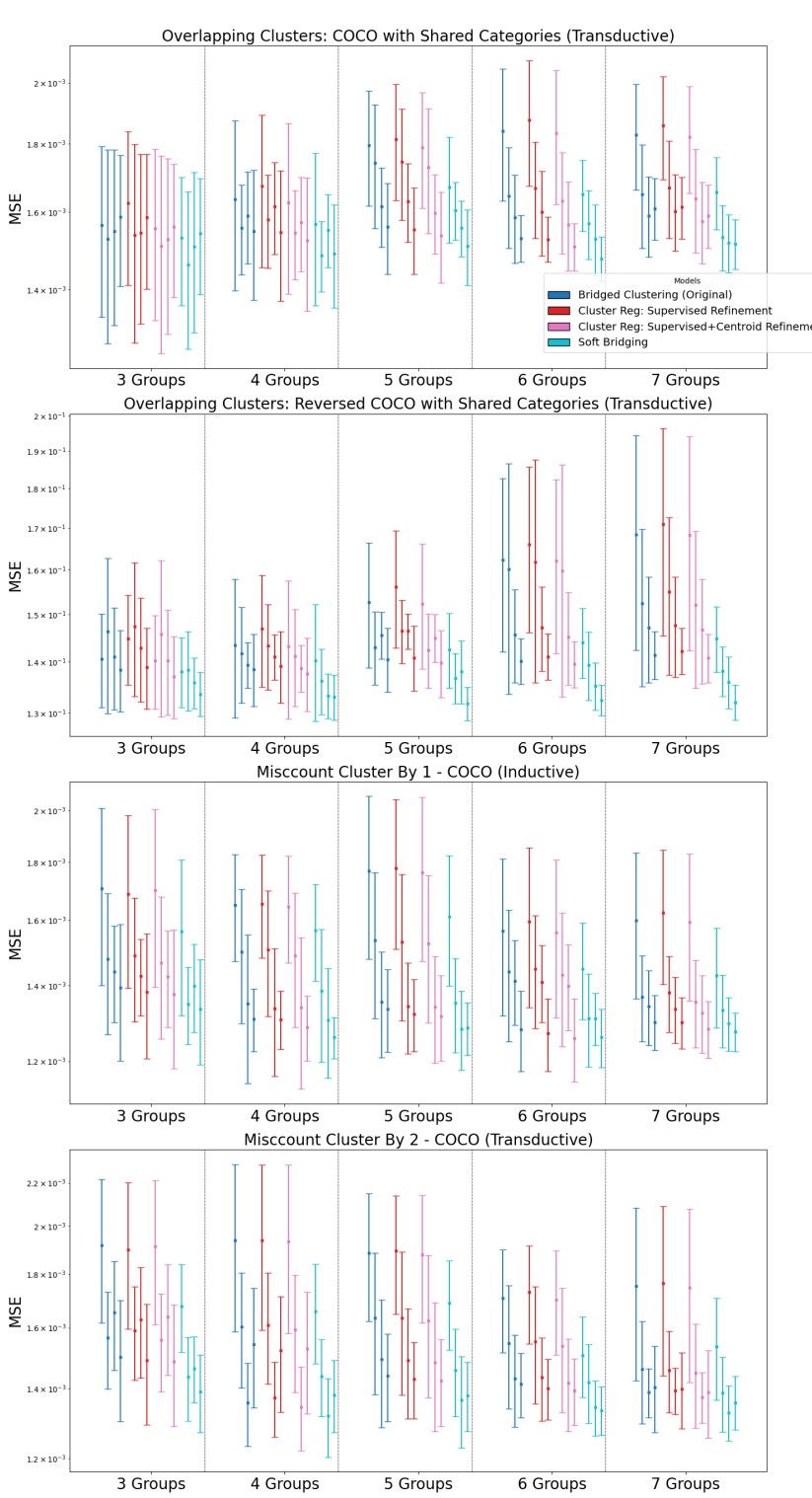

Figure 3: Transductive Experiments using Algorithmic Variants. Similar Observations with Inductive Setting.

# B  EXPERIMENT RESULTS FOR ADVERSARIAL CONDITIONS

Our base model Bridged Clustering does incur more loss in cluster-adversarial conditions, but still maintains overall competitive performance, among the strongest models with low MSE.

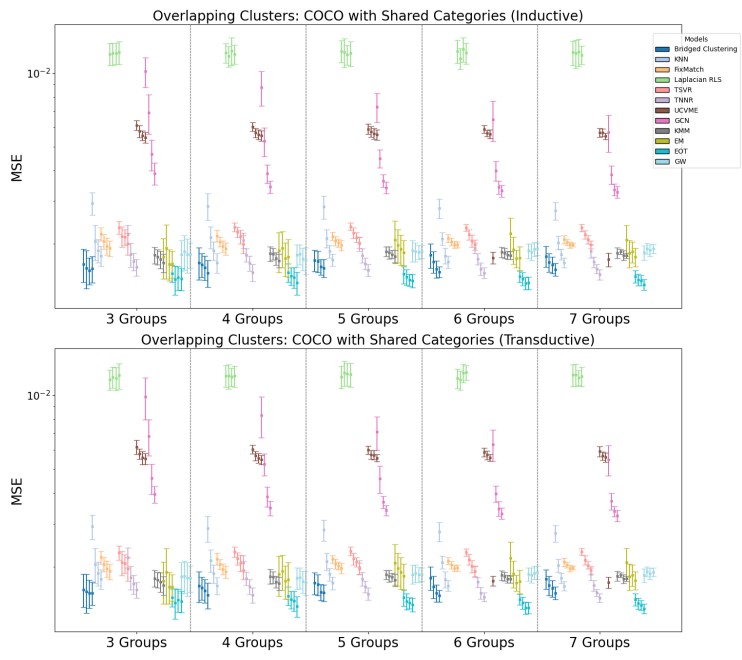

Figure 4: Running main experiment on COCO with Overlapping Data Classes

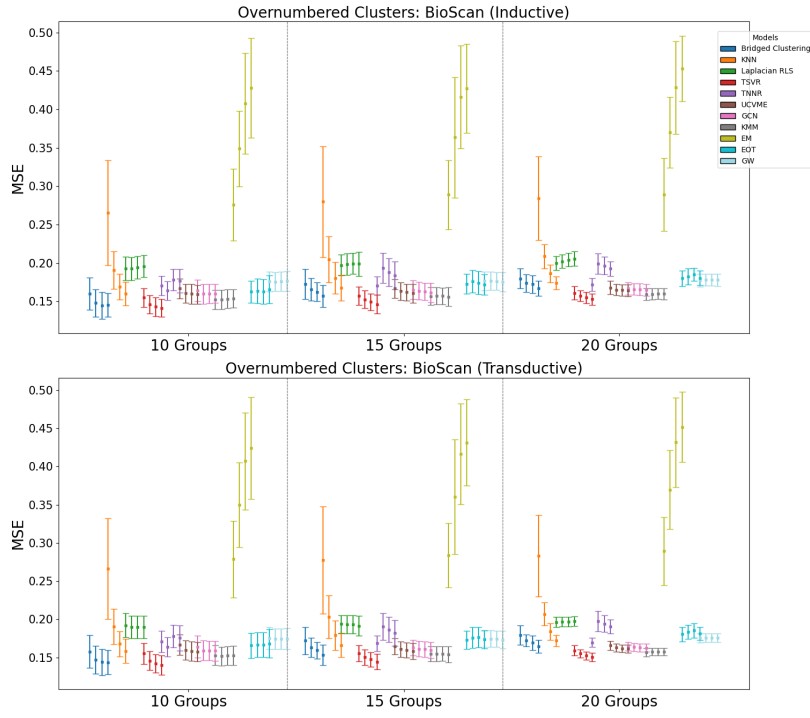

Figure 5: Running main experiment on Large Cluster Counts for Bioscan

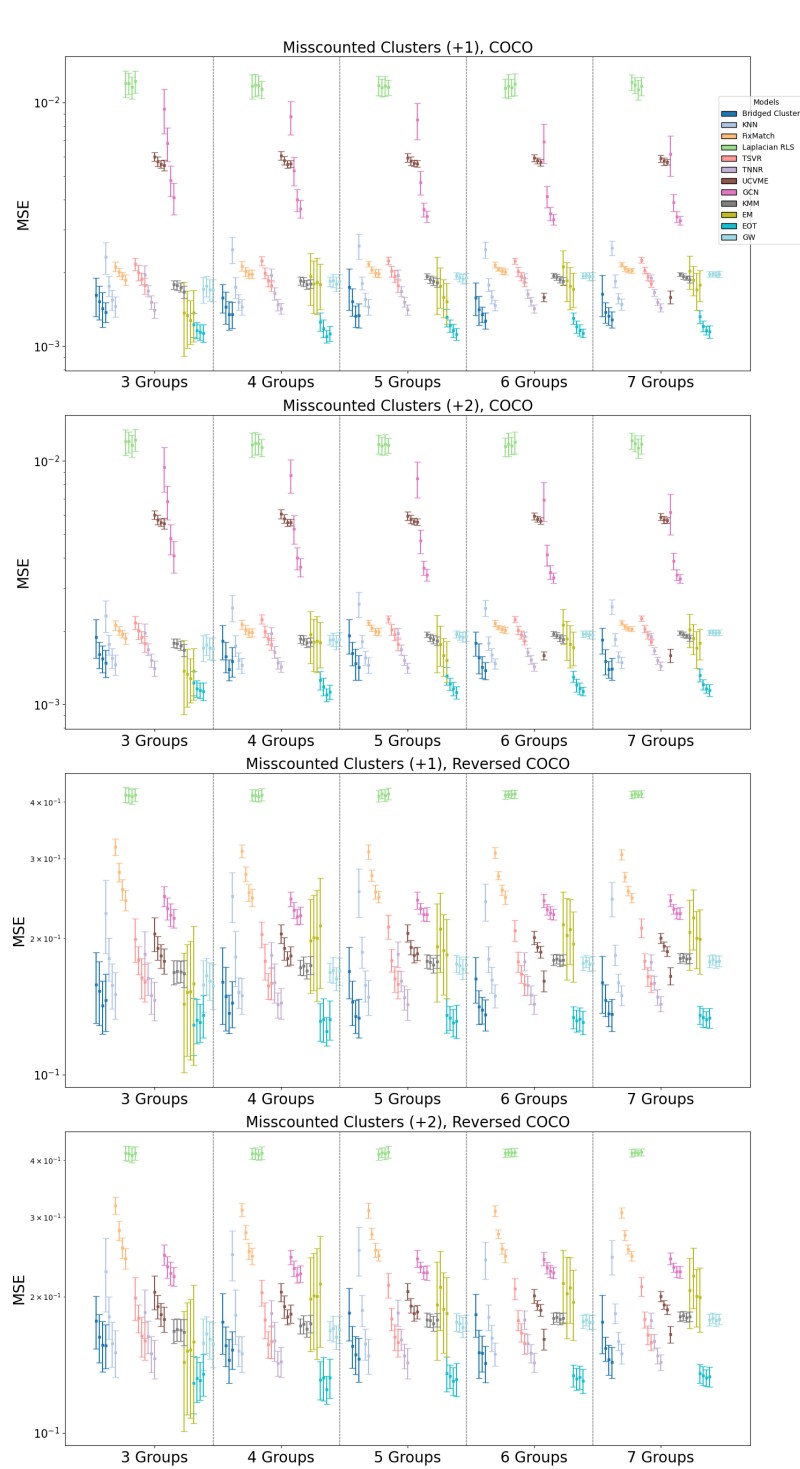

Figure 6: Running main experiment on Misscounted Cluster numbers for COCO and Reversed COCO (Shown inductive results, but transductive outcomes are similar. Miscounting as shown here refers to overcounting the number of clusters, and we note that undercounting experiments also show similar results.)

## C    USING DIFFERENT ENCODERS AND CLUSTERING METHODS

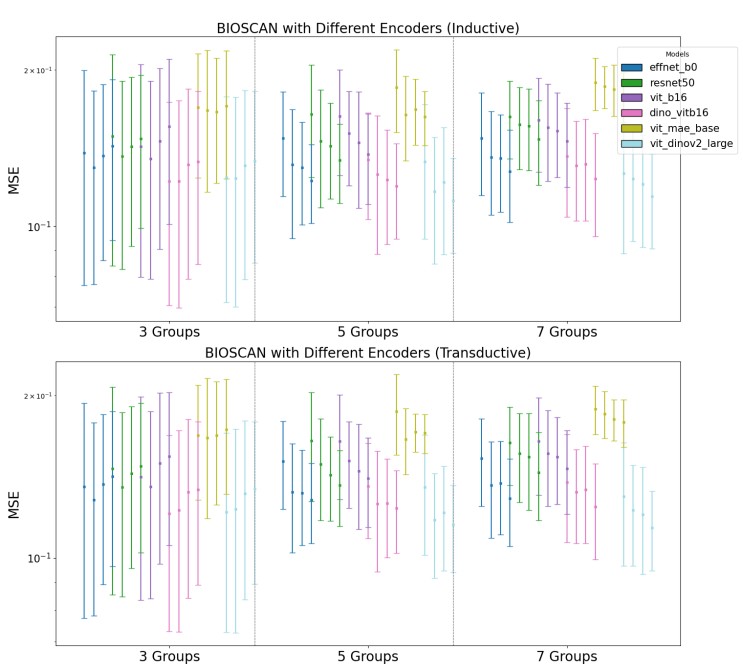

Figure 7: Using Different Image Encoders for Bioscan Experiment

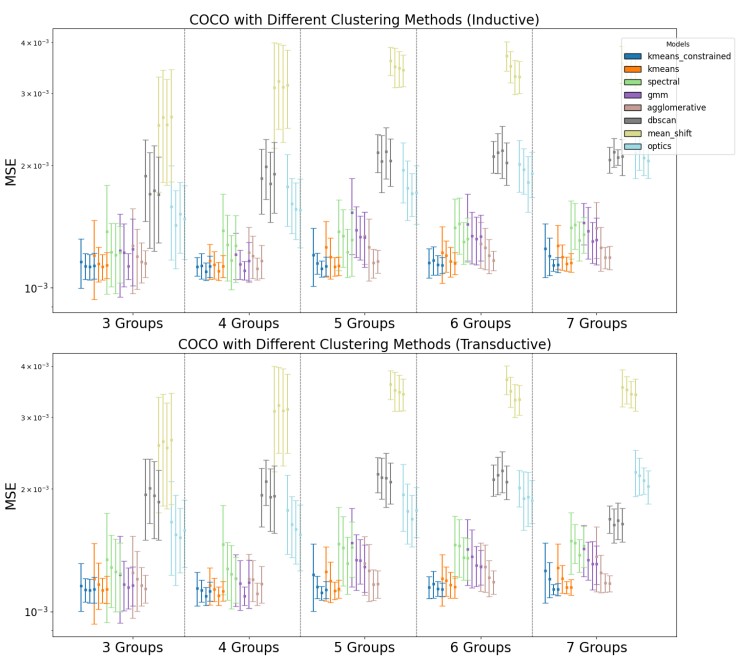

Figure 8: Using Different Clustering Methods for COCO Experiment

## D ILLUSTRATIONS FOR BRIDGED CLUSTERING

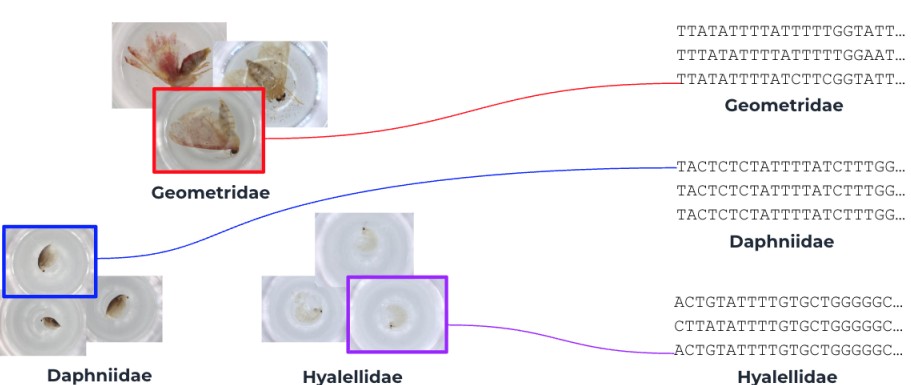

Figure 9: Illustrations for Bridged Clustering in BIOSCAN. In both Image and DNA spaces, datapoints from same species are often clustered together due to embedding proximity. Highlighted datapoints are supervised.

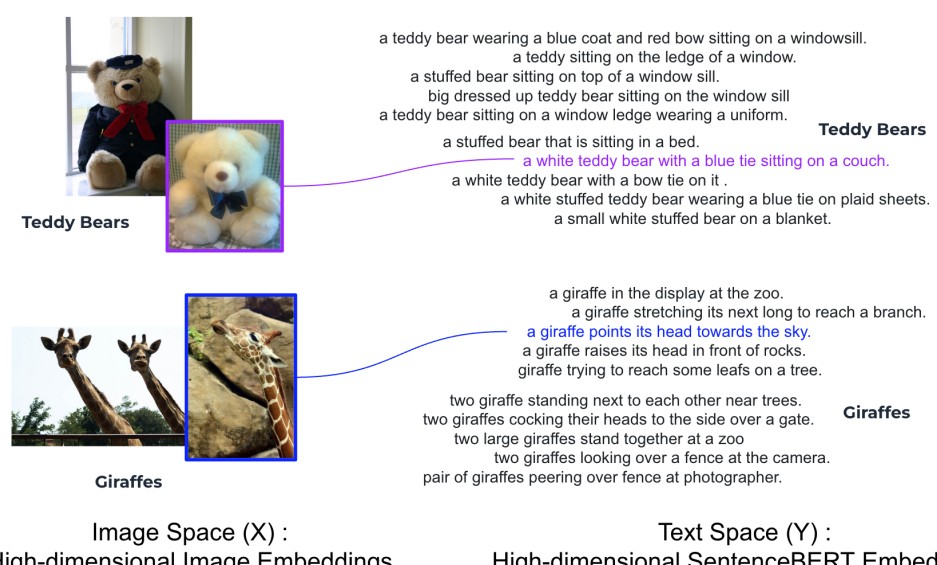

Figure 10: Illustrations for Bridged Clustering in COCO. In both Image and text spaces, datapoints from similar categories are often clustered together due to embedding proximity. Highlighted datapoints are supervised.

# E  BLEU SCORE FOR IMAGE-CAPTIONING EXPERIMENTS

## E.1  BLEU SCORE FOR FLICKR30K EXPERIMENTS (HIGHER IS BETTER)

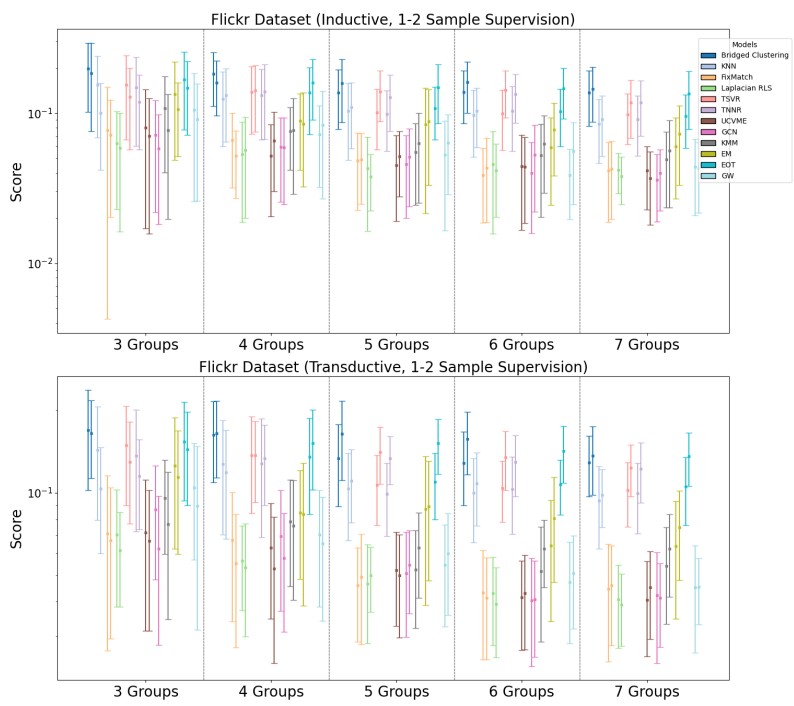

Figure 11: BLEU Score in lower supervision Flickr30k Image-Caption Prediction.

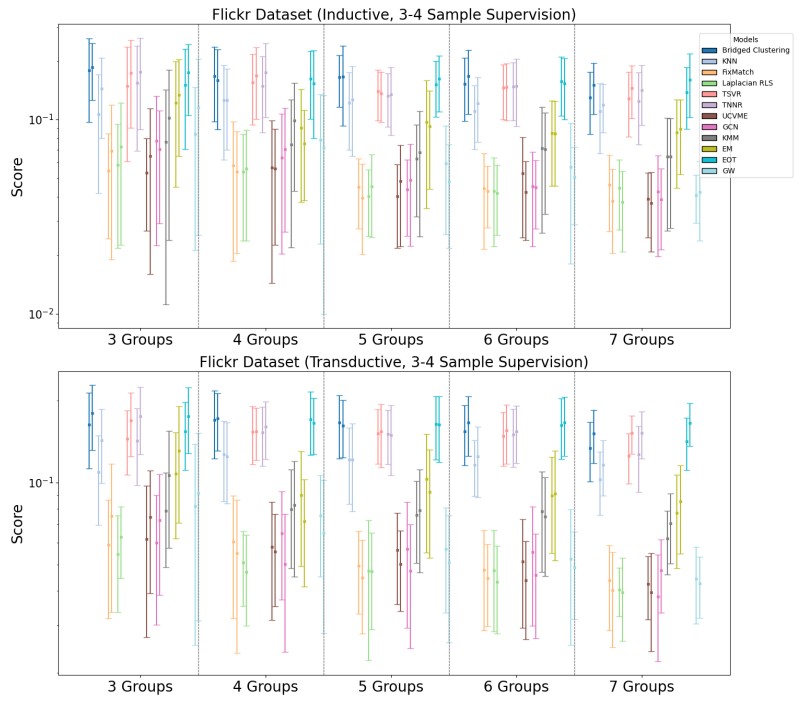

Figure 12: BLEU Score in higher supervision Flickr30k Image-Caption Prediction.

## E.2 BLEU SCORE FOR COCO EXPERIMENTS (HIGHER IS BETTER)

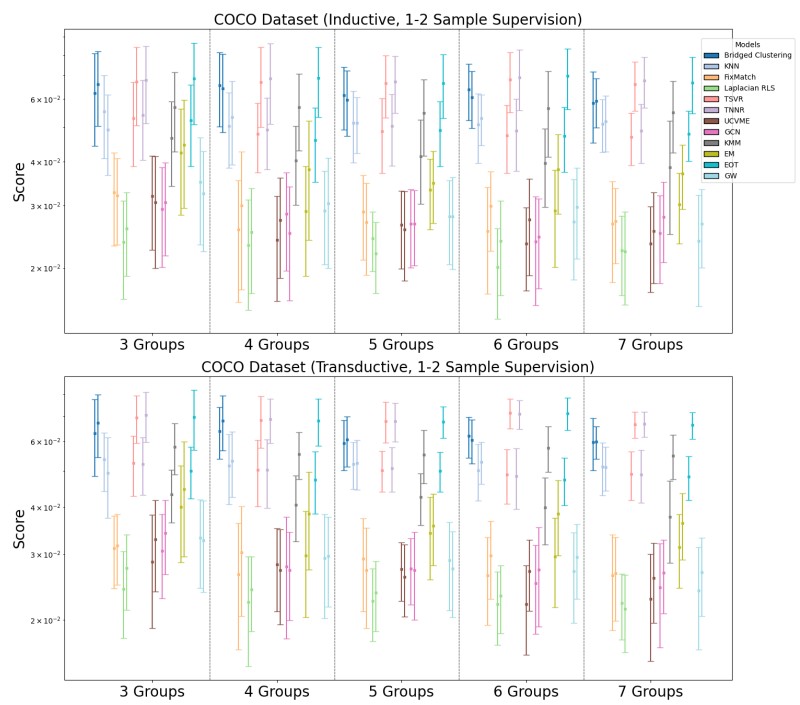

Figure 13: BLEU Score in lower supervision COCO Image-Caption Prediction.

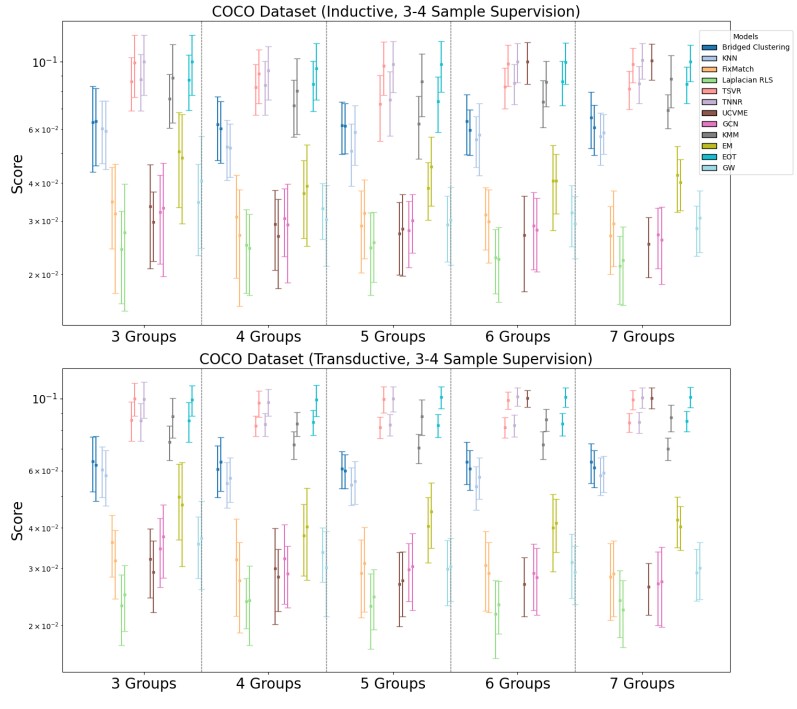

Figure 14: BLEU Score in higher supervision COCO Image-Caption Prediction.

### E.3 BLEU SCORE FOR WIKI EXPERIMENTS(HIGHER IS BETTER)

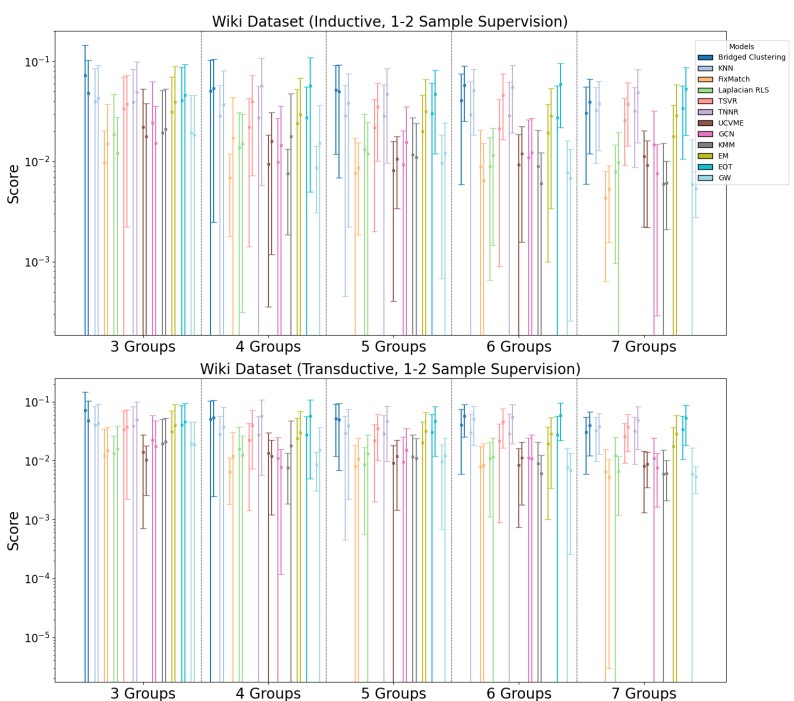

Figure 15: BLEU Score in lower supervision Wiki Image-Caption Prediction.

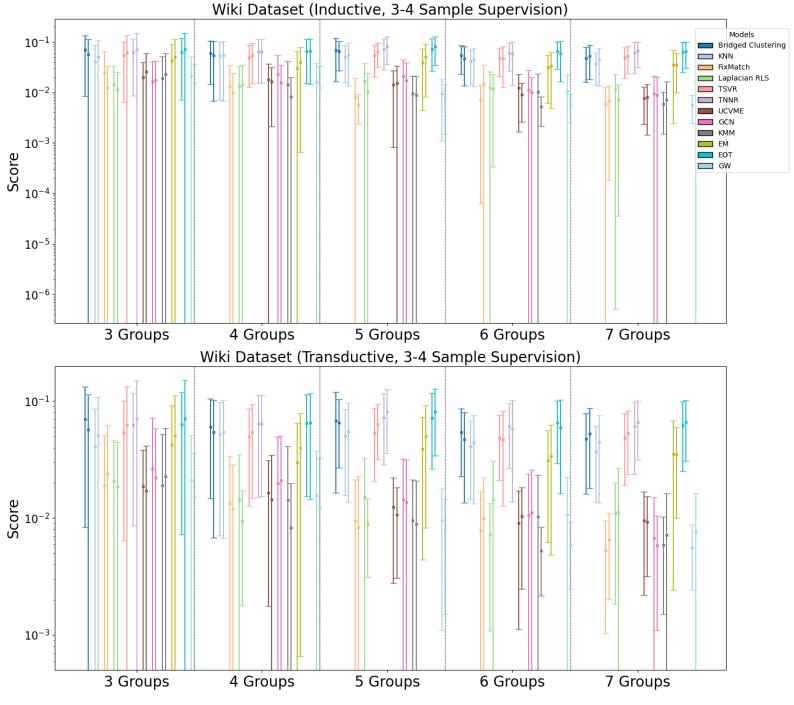

Figure 16: BLEU Score in higher supervision Wiki Image-Caption Prediction.

# F  BRIDGED CLUSTERING RUNTIME

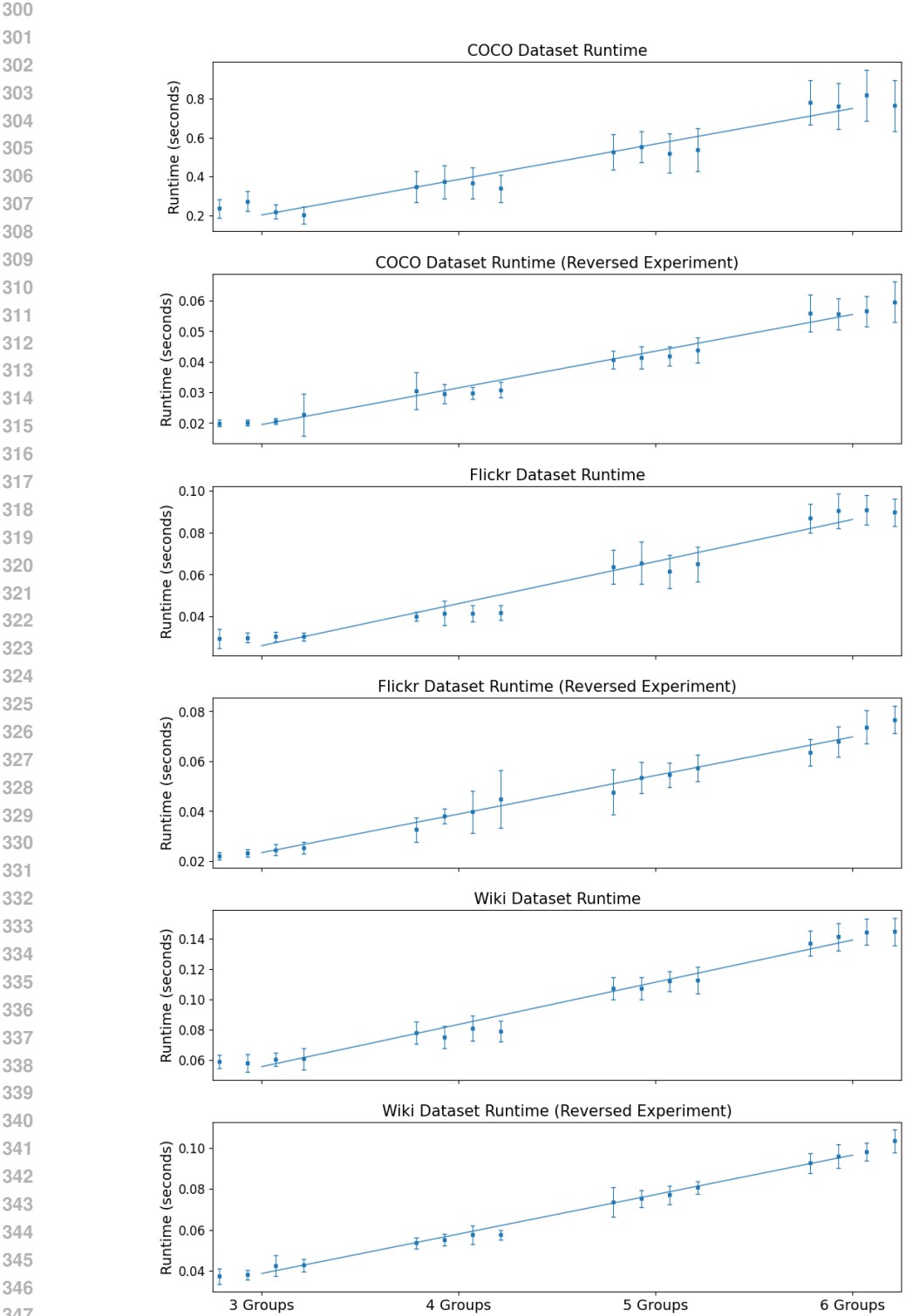

Figure 17: Empirical Runtime shows linear growth, mirroring theoretical bound (Inductive Experiments).

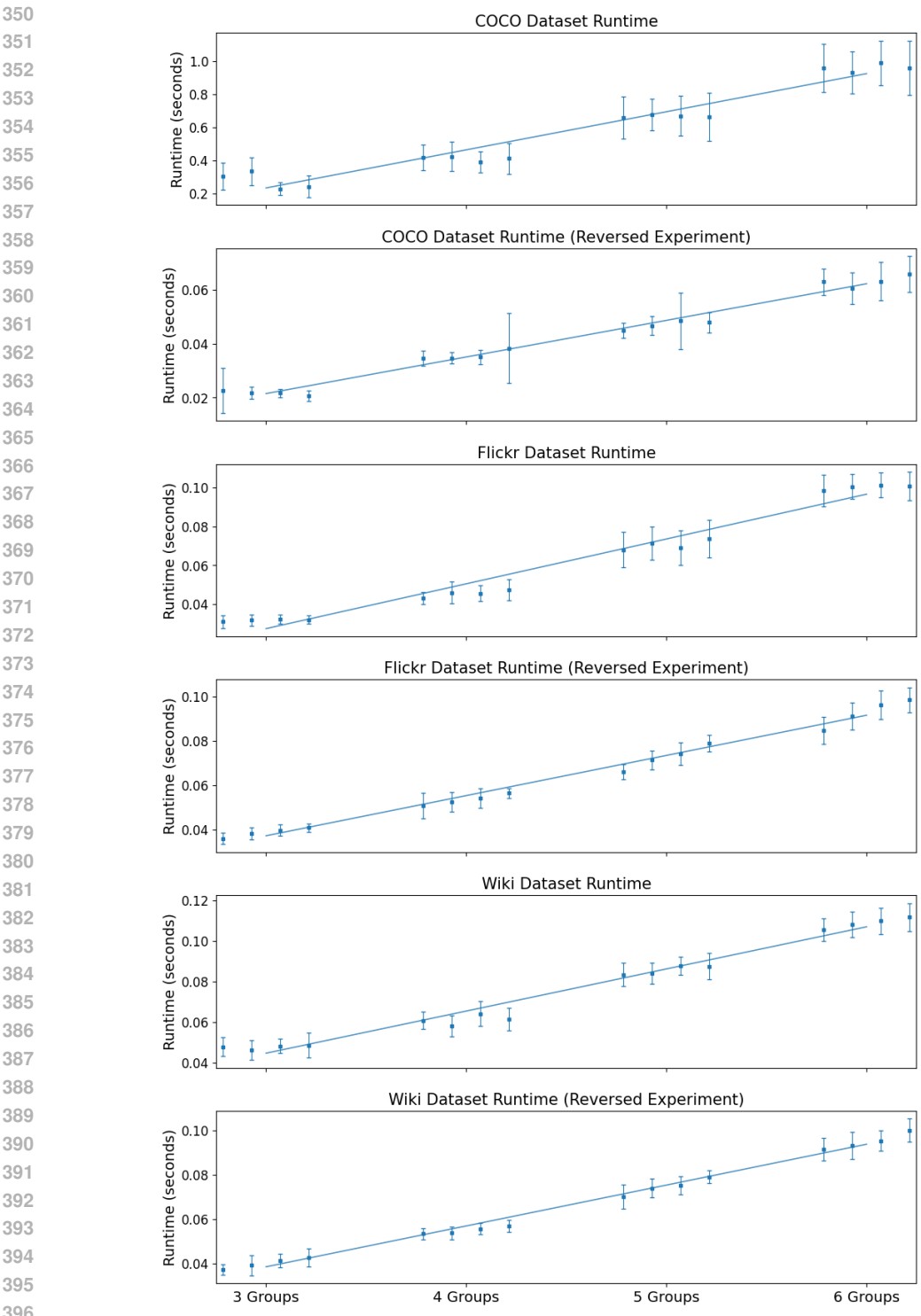

Figure 18: Empirical Runtime shows linear growth, mirroring theoretical bound (Transductive Experiments).

## G    Experiment Details

This appendix details the embedding models, clustering algorithms, and parameters used.[4]

### G.1    BIOSCAN-5M

BIOSCAN-5M contains the specimens images, taxonomic labels, and raw nucleotide barcode sequences of over 5 million insect samples (Gharaee et al., 2024). For experimentation, our models will be given the encoded specimens images ($x$) to predict the encoded nucleotide barcodes ($y$), based on the latent taxonomic labels ($t$). Every sample is labeled according to the hierarchical Linnaean taxonomy: phylum, class, order, family, genus, and species. (Linnaeus, 1789).

We set up our dataset based on the test split of the original BIOSCAN-5M dataset, which is representative of the entire dataset and sizable (39,373) for our purposes. According to the lowest-level taxonomy label of "species", we can divide the dataset into hundreds of homogeneous species groups. Among those species, we randomly choose $C$ groups for each experiment run, prioritizing sampling species with different "family" labels, and then randomly draw 200 samples from each of the selected species groups. By selecting our species groups in this way, we approximately meet our theoretical assumption of sub-Gassian mixtures.

Within each group, samples are randomly assigned according to some fixed ratio to one of the three data-splits: input-only, output-only, and supervised. For consistency, we set the output-only split to be 10%, and vary the supervised percentage, using the rest as input-only data (When testing the inverse direction, we keep the input-only split to be 10%, vary the supervised percentage, and use the rest as output-only data). Pooling all data, we remove the species information from the samples, hoping that our clustering procedures can group our data roughly according to latent species affiliations.

To convert the raw data into cluster-able data that we can easily train and evaluate on, we encode the specimens images with a EfficientNetB0 image encoder (Tan & Le, 2019) and the DNA barcodes with BarcodeBERT (Arias et al., 2023). We also narrow our sampling to the Insecta Class since it contains 98% of the BIOSCAN-5M data.

### G.2    WIT

Wikipedia Image-Text (WIT) (Srinivasan et al., 2021) Dataset is a large curated set of more than 37 million image-text associations extracted from Wikipedia articles. Each entry of the dataset contains an Wikipedia image with its image caption, as well as information of all pages containing this particular image.

To curate our dataset, we select the first 100,000 samples from WIT, encoding the images with ResNet50 (He et al., 2016)and the text captions with Sentence-BERT(Reimers & Gurevych, 2019).

We further extract the English page titles associated with every image and encode them with Sentence-BERT. We use DBSCAN(Schubert et al., 2017) to discover dense clusters of datapoints with neighboring page title encodings, which suggests that these datapoints are associated with a latent category of Wikipedia pages. To further ensure closeness, we add another eligibility requirement for the clusters: every member in the cluster must share one word in their page title. Among these dense clusters retrieved, we randomly sample $C$ clusters, choose 25 samples from each cluster to be included in each run, and split them into three data-splits as in BIOSCAN (input-only split containing only images, output-only split containing only image captions, and the fully supervised split). This time we adjust the output-only percentage to 20%, as the cluster sizes become small and it'll be challenging to form effective output clusters with only 10% data. Pooling all data, we remove the page title information from the samples, hoping that our clustering procedures can group our data roughly according to the latent Wikipedia page categories.

---

[4]The runtime and memory requirements depend heavily on the chosen clustering algorithm; our current implementation is not optimized, which complicates empirical evaluation.

### G.3 FLICKR30K

The Flickr30k dataset (Young et al., 2014) is comprised of 31,783 images that capture people engaged in everyday activities and events. For each image, the dataset provides 5 human-annotated descriptive captions.

We select the first 10,000 samples from Flickr30k, and split each sample into 5 datapoints, so that each entry contains one unique caption and one (possibly repeated) image.

In close parallel to the WIT experiment, we encode the images with ResNet50 and the text captions with Sentence-BERT. To select meaningful families of data, we further encode the text captions with TF-IDF and perform clustering, so that caption that share key words would be clustered close together. We then use DBSCAN to discover dense clusters of datapoints, randomly sample $C$ clusters and sample 25 datapoints per cluster to be included each experiment run. We split the data in each cluster into three data-splits as before, then pool all data for the experiment.

A slight difference from WIT is in caption prediction, where we measure the distance between our model-predicted caption and the closest of the 5 captions that image corresponds to.

### G.4 COCO

The Microsoft Common Objects in Context Dataset (COCO (Lin et al., 2014)) is another image-text dataset. Every image in the dataset is paired with an array of text annotations, as well as a list of categorical labels, corresponding to different objects present in the image.

Our curated dataset is extracted from the validation split of the original COCO dataset. We sample all datapoints with only one categorical label in their list, and group them according to their categorical labels. In each experiment run, we randomly sample C groups and sample 200 datapoints per group, then follow the experimental procedure as exactly described in Flickr30k.

### G.5 NOTE ON BASELINES

We adapted the baseline methods to our experiment environment, keeping faithful to the original methods and ensuring conceptual correctness. We allocated a reasonable fine-tuning budget for our baseline methods with a small grid across key hyper-parameters. Since BRIDGED CLUSTERING does not require any finetuning, we didn't spend excessive resource in the fine-tuning of the baselines, so the baselines used in our experiments may not be the optimized version. For baseline methods that involve iterations, we set a reasonable and generous maximum iteration according to the runtime of each algorithm, which is normally 2000, as compared to 100 or less maximum iterations for BRIDGED CLUSTERING. Please see our code for additional details. We also attach the tuning grid for reference.

## H ON THE IMPORTANCE OF REPRESENTATION OR BRIDGED CLUSTERING

Following our algorithm, we independently cluster the input space (e.g., encoded images) and the output space (e.g., encoded DNA or text). In expectation, effective clustering shall retrieve latent class affiliations. For example, in the Bioscan experiment where the latent variable is "species", samples assigned to the same input cluster shall have the same species label, as with output cluster. We measure this clustering effectiveness using Adjusted Mutual Information (AMI) between cluster assignments and latent labels for both inputs and outputs.

Since our algorithm is model-agnostic, we are free to experiment with a wide range of clustering methods. As we know the number of latent variables included in every experiment, hence the ideal number of clusters, k-clustering methods are natural choices: k-means, spectral clustering, and gaussian mixture model. We also explored agglomerative clustering as a baseline. As we discovered over iterative testing, constrained k-means with balanced cluster sizes is our best performer as measured through AMI, possibly due to the fact that we curated our dataset with equal species-cluster sizes.

Another metric that's directly related to the algorithm's effectiveness is the accuracy of bridged association learning. After clustering, as each supervised point is associated with one input and one

Table 2: Hyperparameter grids used for baseline fine-tuning. Each baseline has a small grid over key hyperparameters; BRIDGED CLUSTERING and KNN require no tuning. Iterative methods use a generous cap (typically 2000 iterations) versus $\leq 100$ for BRIDGED CLUSTERING.

| Method | Grid |
|---|---|
| FixMatch | `lr={1e-4, 3e-4, 1e-3}; batch_size={32, 64}; alpha_ema={0.99, 0.999}; lambda_u_max={0.5, 1.0}; rampup_length={10, 30}; conf_threshold={0.05, 0.1}` |
| LapRLS | `lam={1e-5, 1e-3, 1e-1}; gamma={1e-3, 1e-1, 1}; k={5, 10, 20}; sigma={0.5, 1.0, 2.0}` |
| TNNR | `rep_dim={32, 64, 128}; beta={0.01, 0.1, 1.0}; lr={1e-4, 3e-4, 1e-3}` |
| TSVR | `C={0.1, 1, 10}; epsilon={0.01, 0.1}; gamma={0.1, 1}; self_training_frac={0.1, 0.2, 0.5}` |
| UCVME | `lr={1e-4, 3e-4, 1e-3}; w_unl={1, 5, 10}; mc_T={5, 10}` |
| GCN | `hidden={32, 64, 128}; dropout={0.0, 0.1, 0.3}; lr={1e-3, 3e-3}` |
| KMM | `alpha={1e-2, 1e-1}; kmm_B={100, 1000}; kmm_eps={1e-3, 1e-2}; sigma={0.5, 1.0}` |
| EM | `n_components={2, 3}; max_iter={100, 200}; tol={1e-3, 1e-4}; eps={1e-3, 1e-4}` |
| EOT | `eps={1e-3, 1e-2, 1e-1, 1, 10}; ridge_alpha={1e-2, 1e-3, 1e-4}; tol={1e-5, 1e-7, 1e-9}` |
| GW | `max_iter={200, 400, 800}; tol={1e-5, 1e-7, 1e-9}` |

output cluster, we obtain a voted mapping between input and output clusters. We measure the accuracy of this mapping by comparing it to the ground-truth. Since every input/output cluster roughly corresponds to one distinct latent variable, with high probability every latent variable will dominate exactly one input cluster and one output cluster, allowing us to obtain a non-overlapping ground-truth mapping through latent-by-latent reconstruction. After trials with different voting mechanisms, including Margin-based voting and Hungarian Algorithm, the most accurate mechanism turns out to be simple majority-based voting.

## I  BRIDGED CLUSTERING ALGORITHM (FULL VERSION WITH INVERSE PREDICTION)

This is an alternative version of our algorithm that would be potentially useful in practice, using the bidirectional property of Bridged Clustering.

---

**Algorithm 2** BRIDGED CLUSTERING Algorithm (Full version with Inverse Prediction)

---

**Input**: Input-only set $\mathcal{X} = \{x_1, x_2, \ldots, x_{|\mathcal{X}|}\}$; Output-only set $\mathcal{Y} = \{y_1, y_2, \ldots, y_{|\mathcal{Y}|}\}$; sparse supervised set $\mathcal{S} = \{(x'_i, y'_i)\}_{i=1}^{k}$; Input Test-set $\mathcal{X}_{\text{test}} = \{x_1, x_2, \ldots, x_{|\mathcal{X}_{\text{test}}|}\}$; Output Test-set $\mathcal{Y}_{\text{test}} = \{y_1, y_2, \ldots, y_{|\mathcal{Y}_{\text{test}}|}\}$ (Note: $\mathcal{X}_{\text{test}} = \mathcal{X}$, $\mathcal{Y}_{\text{test}} = \mathcal{Y}$ in transductive setting)

**Output**: $\hat{\mathcal{Y}} = \{\hat{y}_1, \hat{y}_2, \ldots, \hat{y}_{|\mathcal{X}_{\text{test}}|}\}$, $\hat{\mathcal{X}} = \{\hat{x}_1, \hat{x}_2, \ldots, \hat{x}_{|\mathcal{Y}_{\text{test}}|}\}$

1: **Call** Algorithm 1
2: Using the supervised set $\mathcal{S}$, learn the reversed mapping $\hat{A}_{y \to x}$ between clusters in $\mathcal{Y}$ and clusters in $\mathcal{X}$.
3: **for** each sample $y_i$ in $\mathcal{Y}_{\text{test}}$ **do**
4:     Assign $y_i$ to a cluster $c_y = \mathcal{C}_{\mathcal{Y}_{\text{test}}}(y_i)$.
5:     Find corresponding cluster in $\mathcal{X}$: $c_x = \hat{A}_{y \to x}(c_y)$.
6:     Predict $\hat{x}_i$ as the centroid of cluster $c_x$ in $\mathcal{X}$.
7: **end for**
8: **return** Predicted outputs $\hat{\mathcal{Y}}$ for given inputs, Predicted inputs $\hat{\mathcal{X}}$ for given outputs.

---

# J  DETAILED EXPERIMENT RESULTS: MSE

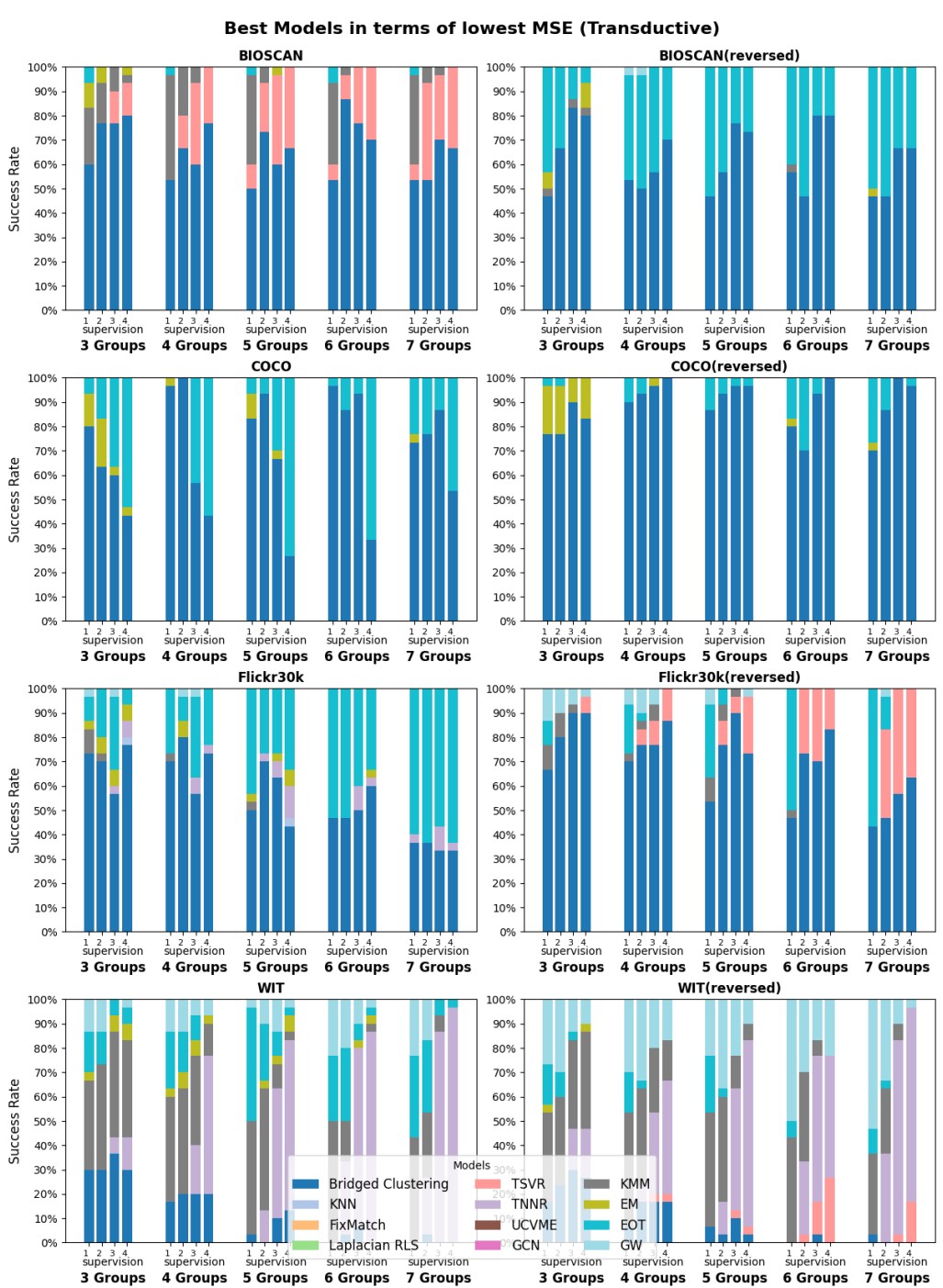

Figure 19: Tansductive Experiment: Best models in terms of lowest MSE, computed across 30 randomized trials per setting. Each bar represents the 30 trials of one setting. For example, if BRIDGED CLUSTERING achieves the lowest MSE among all models in 15 out of the 30 trials for some setting, the bar that corresponds to that setting will be colored 50% blue. The 1,2,3,4 ticks on the bottom represent the settings with 1,2,3,4 supervised samples per cluster.

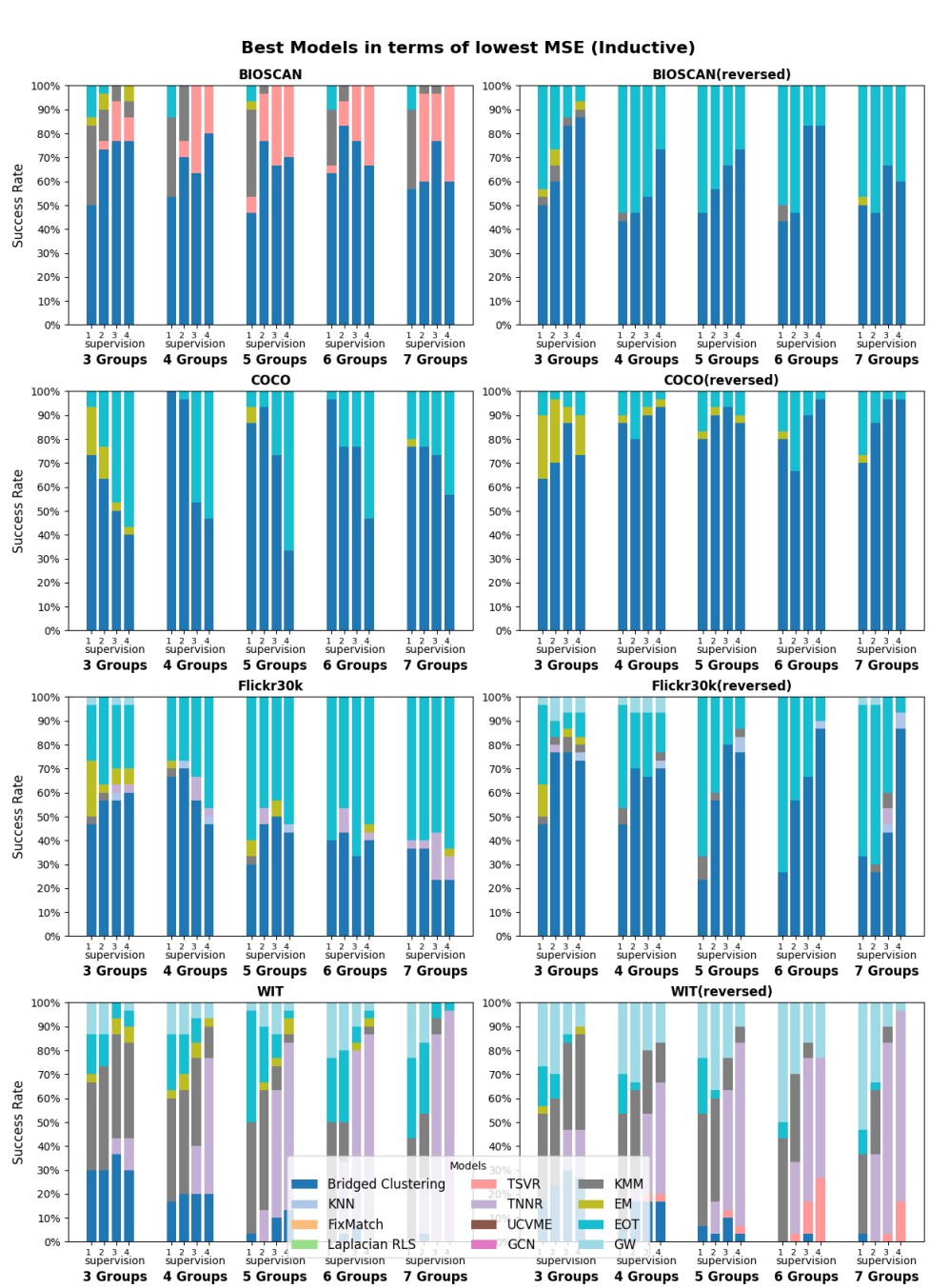

Figure 20: Inductive Experiment: Best models in terms of lowest MSE, computed across 30 randomized trials per setting. Each bar represents the 30 trials of one setting. For example, if BRIDGED CLUSTERING achieves the lowest MSE among all models in 15 out of the 30 trials for some setting, the bar that corresponds to that setting will be colored 50% blue. The 1,2,3,4 ticks on the bottom represent the settings with 1,2,3,4 supervised samples per cluster.

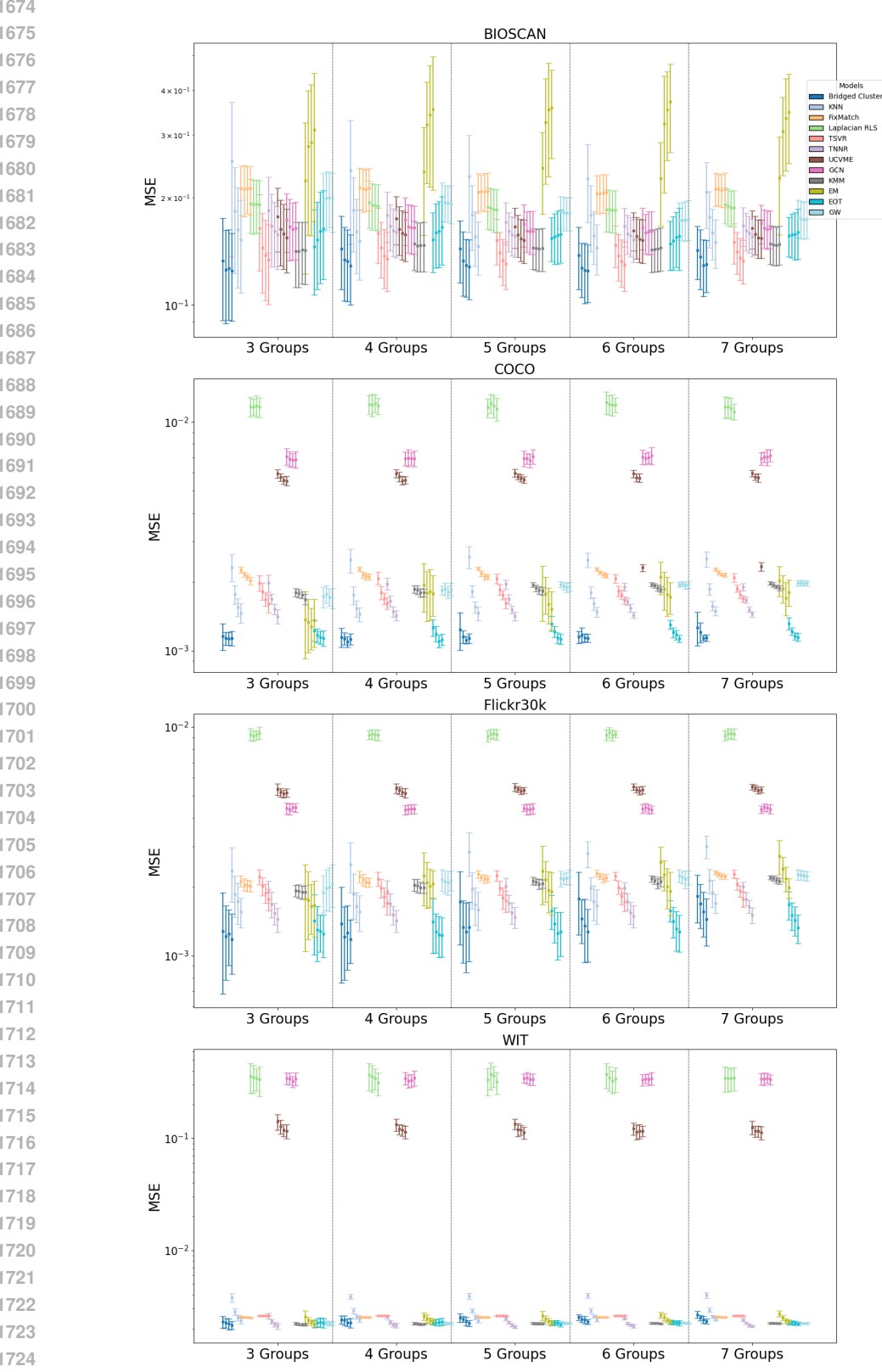

Figure 21: MSE distribution of different models in the transductive setting. The 4 distribution plots of the same color represent the settings with 1,2,3,4 supervised samples per cluster.

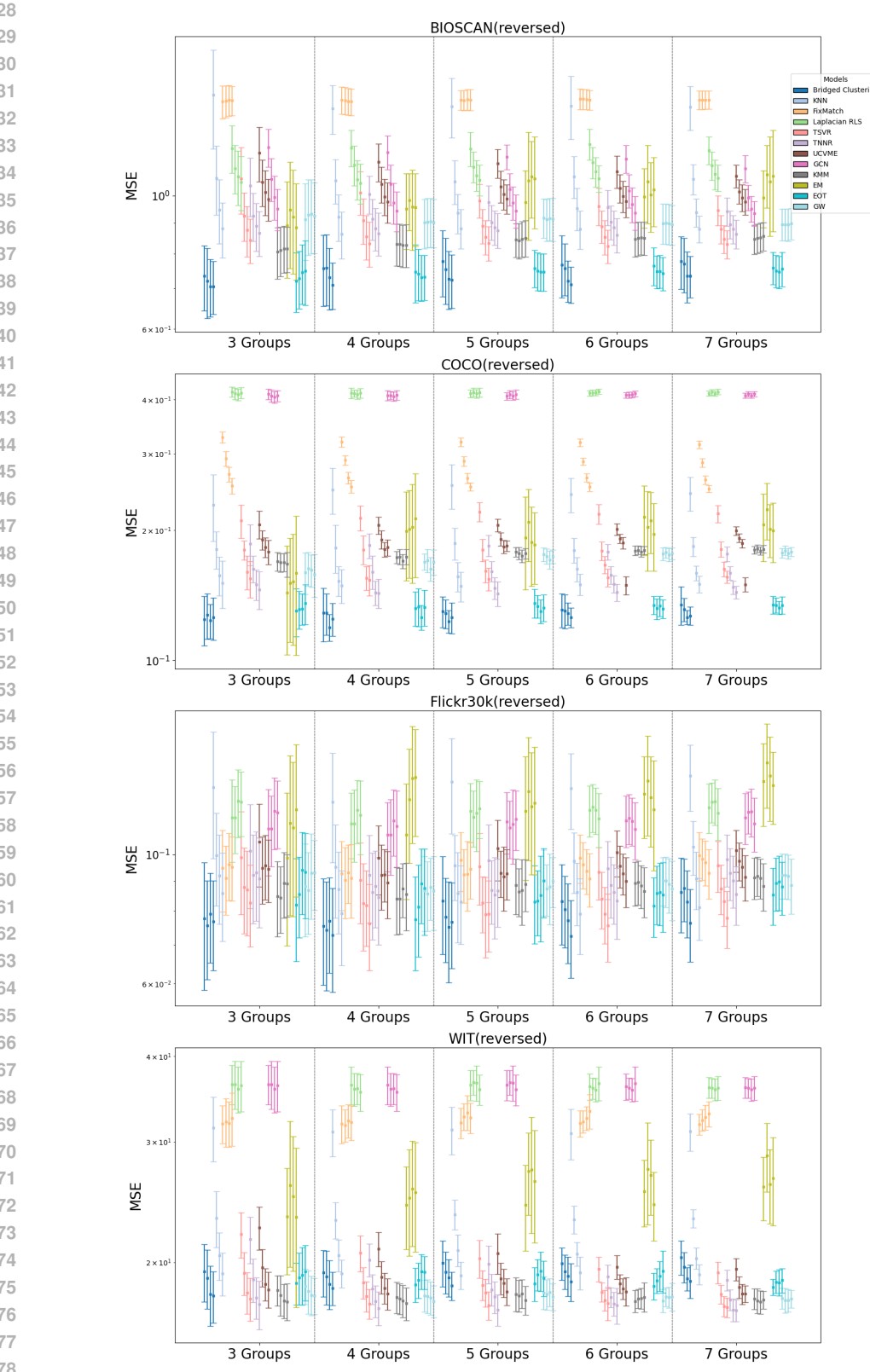

Figure 22: MSE distribution of different models in reversed experiments in the transductive setting. The 4 distribution plots of the same color represent the settings with 1,2,3,4 supervised samples per cluster.

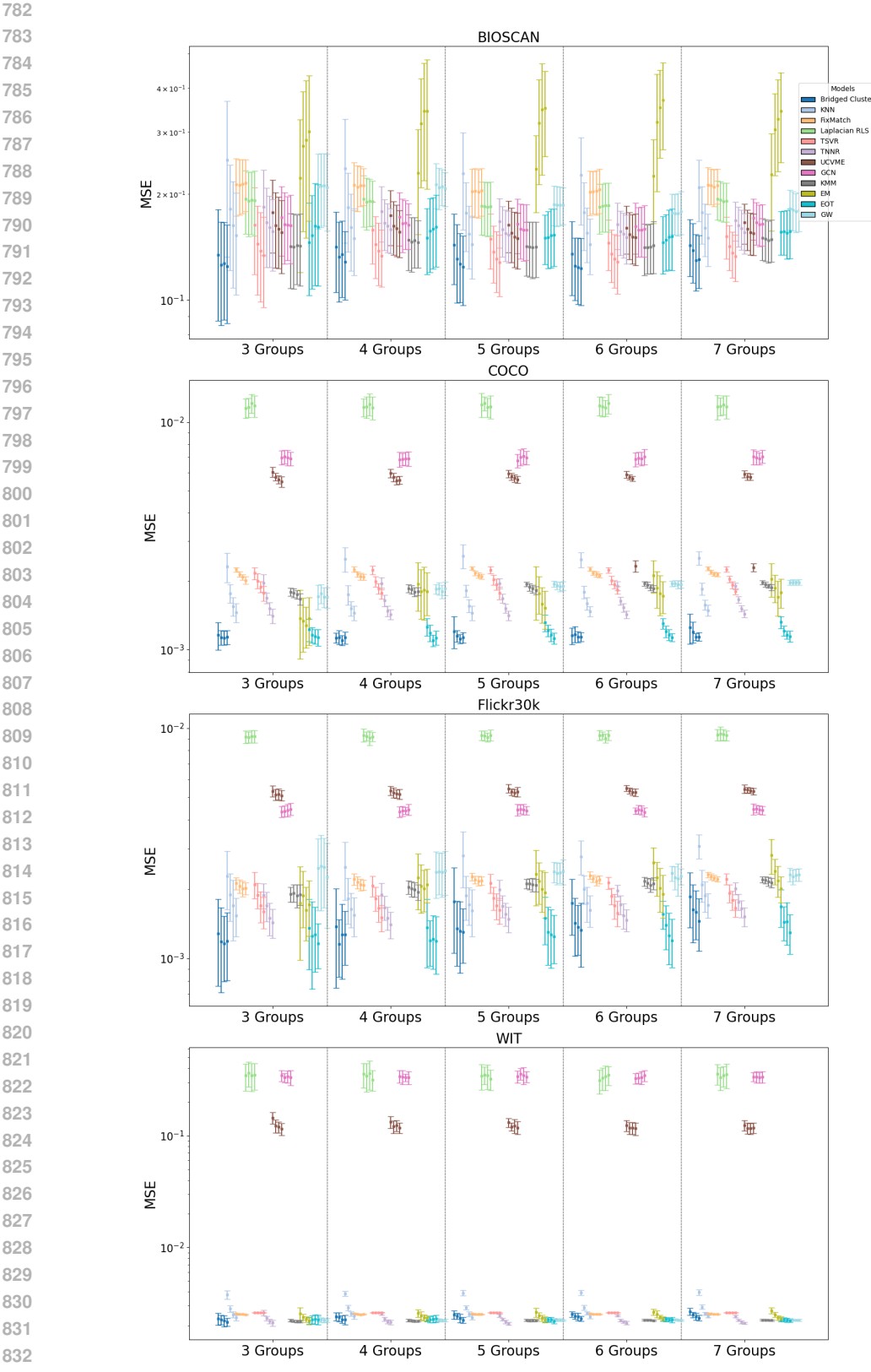

Figure 23: MSE distribution of different models in the inductive setting. The 4 distribution plots of the same color represent the settings with 1,2,3,4 supervised samples per cluster.

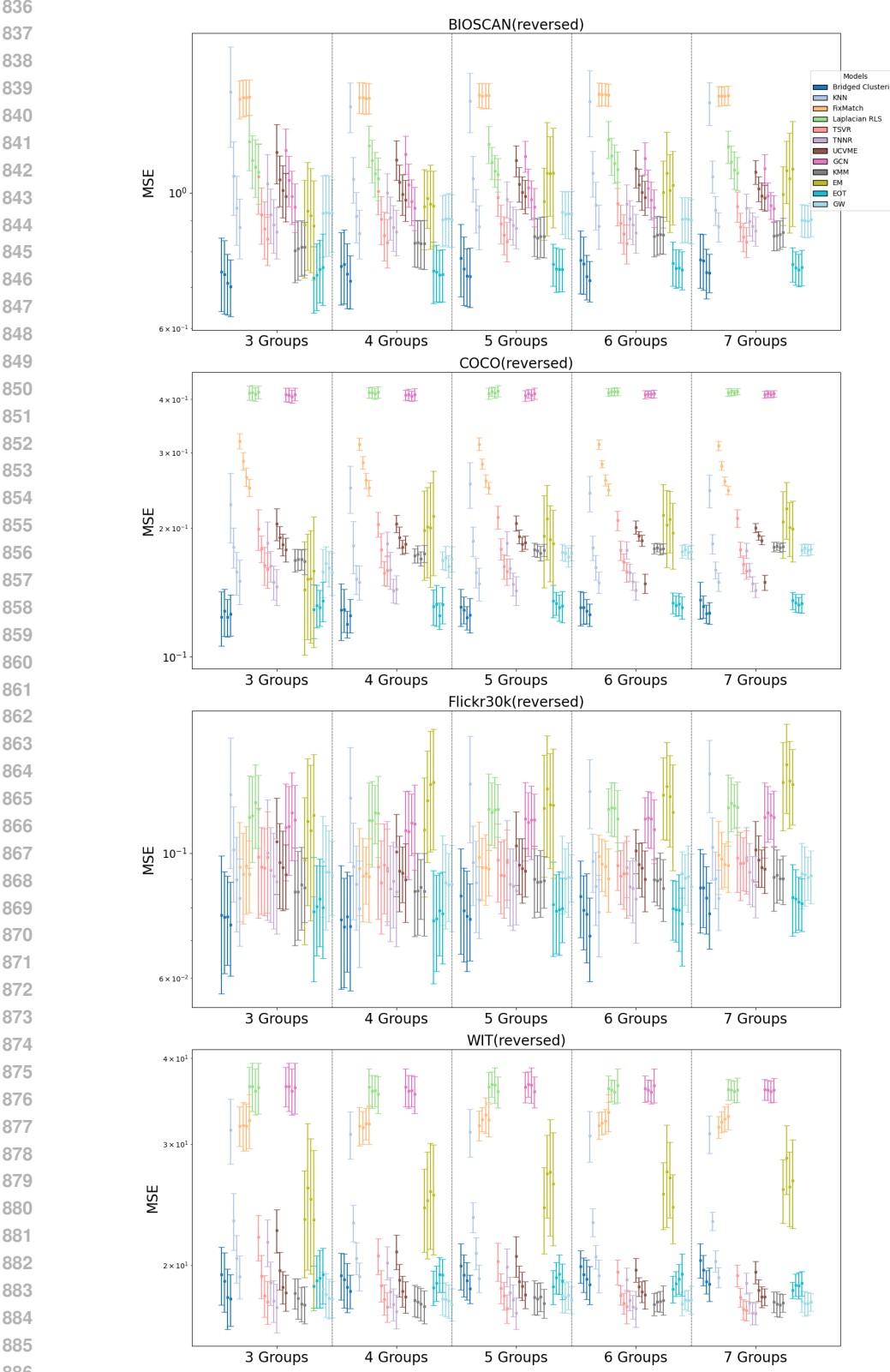

Figure 24: MSE distribution of different models in reversed experiments in the inductive setting. The 4 distribution plots of the same color represent the settings with 1,2,3,4 supervised samples per cluster.

# K DETAILED EXPERIMENT RESULTS: CLUSTER QUALITY AND ACCURACY (TRANSDUCTIVE)

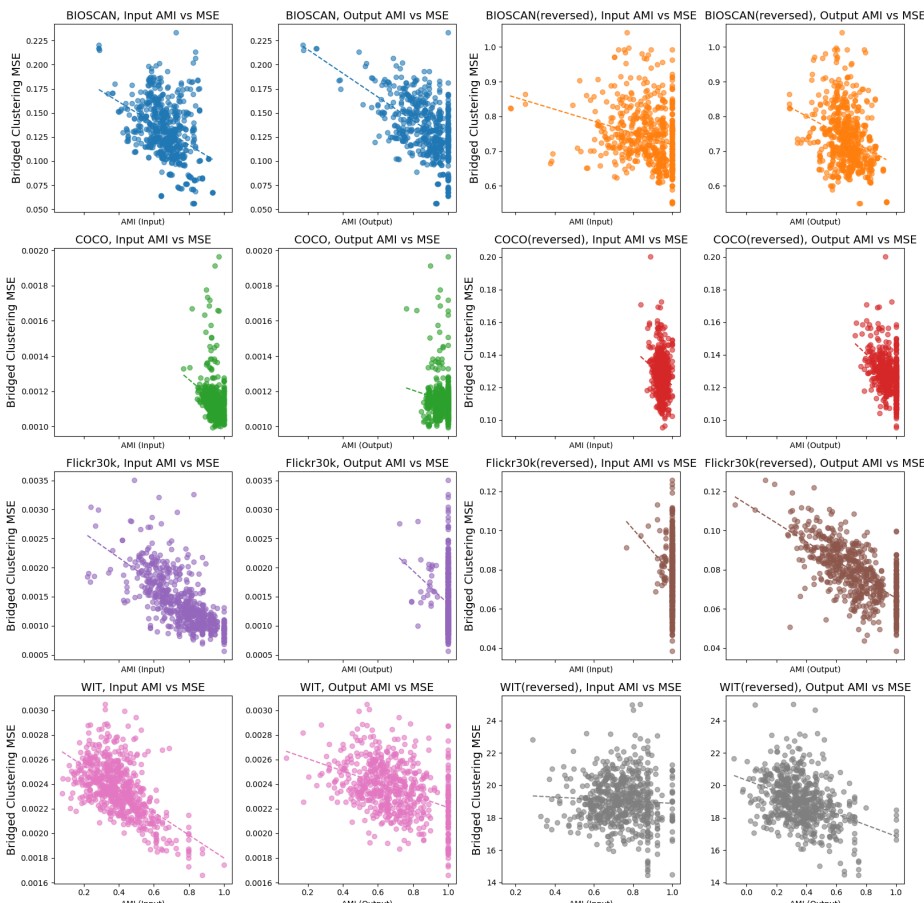

Figure 25: Mean Squared Error of BRIDGED CLUSTERING V.S. Adjusted Mutual Information of Clusters.

# L LLM USAGE

We used ChatGPT to aid and polish this paper, including grammatical and writing quality checks.

# M MORE INFORMATION

For the experiments, we use a machine with 48 cores, 516GB RAM, NVIDIA RTX 6000 Ada Generation 49.1GB; no special hardware is required.

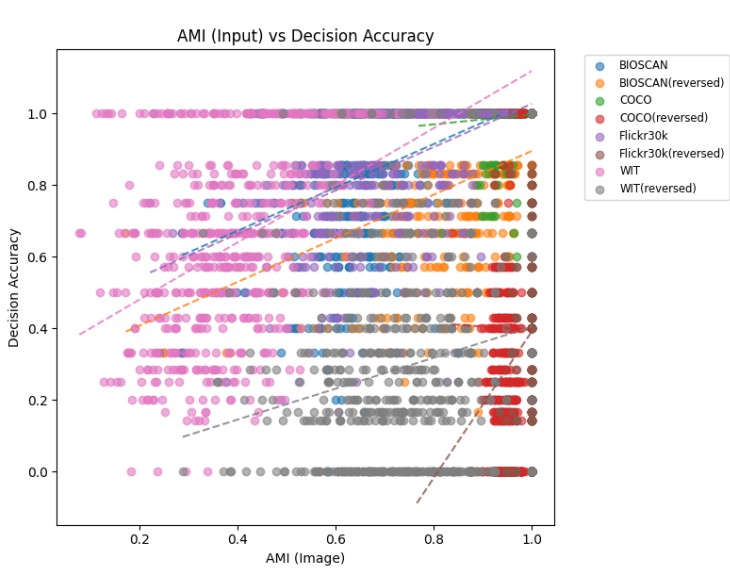

Figure 26: Bridging Accuracy as function of Adjusted Mutual Information of input clusters.

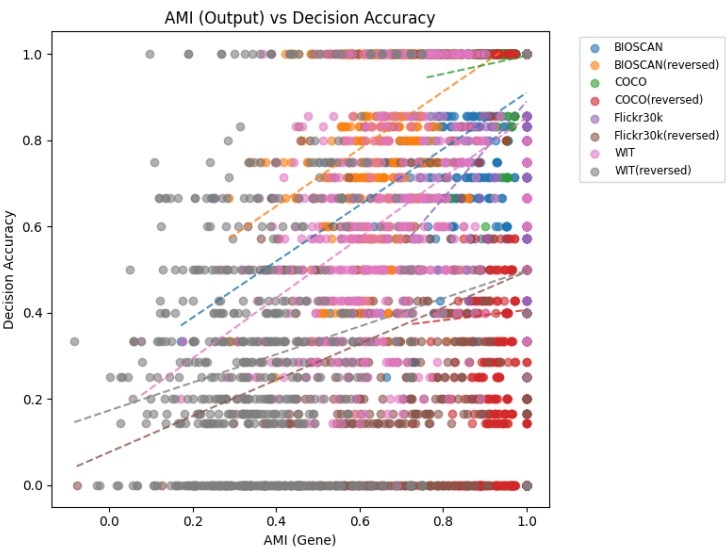

Figure 27: Bridging Accuracy as function of Adjusted Mutual Information of output clusters.

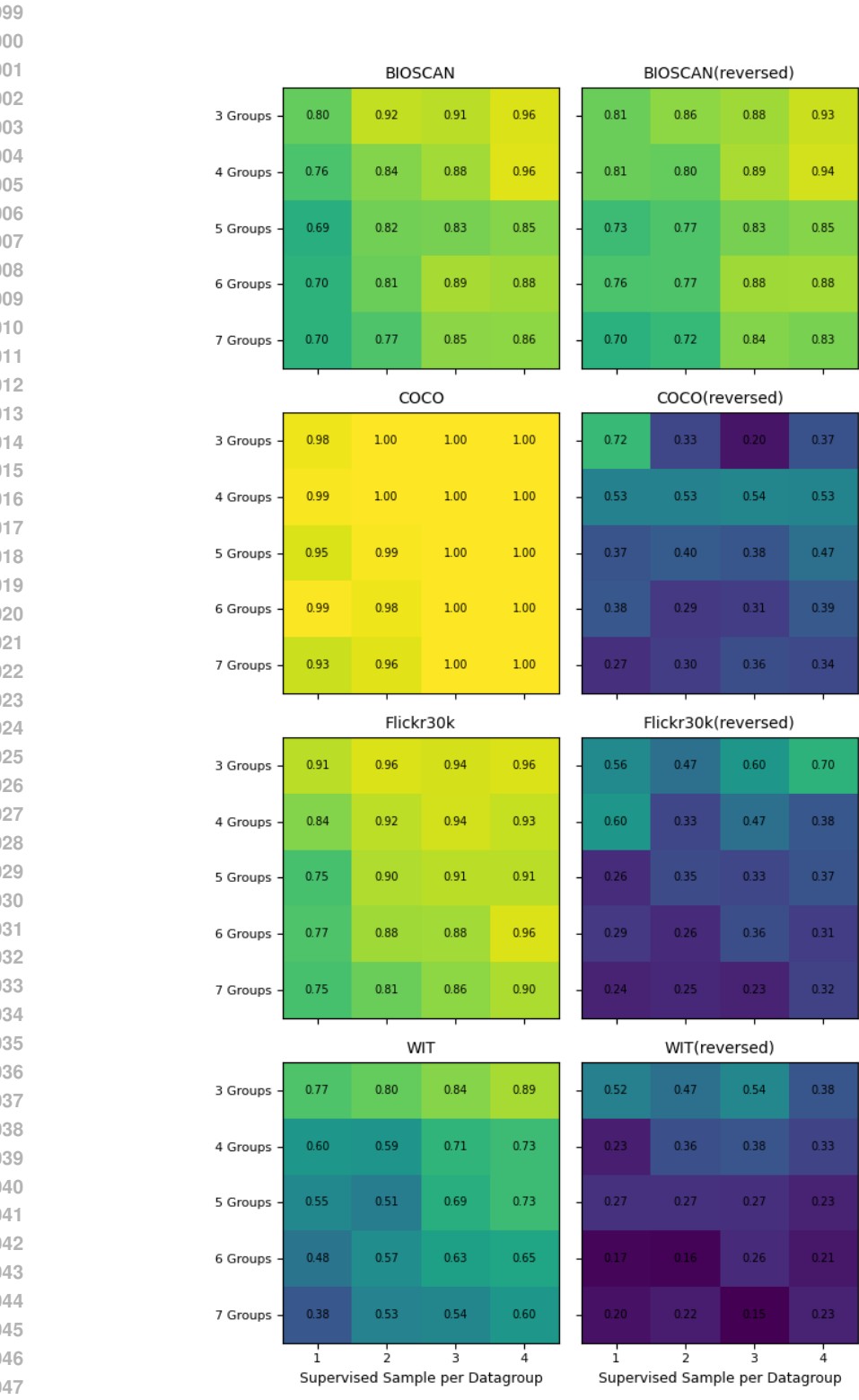

Figure 28: Average Bridging Accuracy.

