# OpenReview forum: "Bridged Clustering for Representation Learning: Semi-Supervised Sparse Bridging"
_ICLR.cc/2026/Conference — Submitted to ICLR 2026_

### Official Review · Reviewer_KEPX · 2025-10-26

**Soundness:** 3
**Presentation:** 2
**Contribution:** 2
**Rating:** 4
**Confidence:** 2

**Summary:**

This paper introduces Bridged Clustering, a semi-supervised learning framework designed to learn predictors from unpaired input and output datasets. The method first clusters the input space and output space independently, then learns a sparse, interpretable bridge between clusters using a small set of paired examples. At inference, a new input is assigned to its nearest input cluster, and the centroid of the linked output cluster is returned as the prediction. The approach is model-agnostic, computationally efficient, and supports bidirectional inference. Theoretical analysis shows that the predictor's risk is bounded under certain mis-clustering and mis-bridging rates. Empirical results on multimodal datasets (BIOSCAN, WIT, Flickr30k, COCO) demonstrate competitive performance against a wide range of baselines, especially in low-supervision regimes.

**Strengths:**

1. The proposed BC is model-agnostic and easy to implement, relying on standard clustering and majority-vote bridging. Its sparse, interpretable mapping between input and output clusters offers transparency absent in dense transport or deep generative models.
2. Evaluation across multiple modalities (vision, language, genomics) under both inductive and transductive settings, along with comparisons to a wide range of baselines, demonstrates robustness and generality.

**Weaknesses:**

1. The method assumes that input and output spaces can be cleanly partitioned into clusters that align one-to-one. This may not hold in real-world data with overlapping or hierarchical categories. When clusters poorly capture latent structure (e.g., in the WIT dataset), predictive performance degrades notably. This limits robustness in high-dimensional or weakly separable spaces.
2. Although the method’s interpretability is emphasized, no concrete visualization or case study (e.g., cluster bridges on BIOSCAN or COCO) is provided to illustrate interpretability in practice.
3. Predictions are always the centroid of the output cluster, which may be too coarse for tasks requiring fine-grained outputs. The model does not refine predictions with additional supervision beyond the bridge.

**Questions:**

1. Since the method heavily relies on representation separability (∆X, ∆Y), how sensitive is performance to pretrained encoder choice? Have the authors experimented with self-supervised embeddings?
2. Empirically, the paper reports success with as few as one labeled pair per cluster. Is there a theoretical lower bound on the number of pairs required to guarantee reliable bridging under bounded εX, εY?

---

> ### Author Response · Authors · 2025-11-22
> **Response to Reviewer KEPX (Part 1)**
>
> Thank you so much for your thoughtful and inspiring review. Your comments on robustness, intra-cluster granularity, interpretability, and representation dependence really resonated with us, and they helped shape substantial improvements to the paper. Addressing your feedback, we present a new Section 8 in our paper, **“Algorithmic Variants and Extended Applicability”**. We expanded our experiments with explicit adversarial settings, designed robustness variants of Bridged Clustering that performed strongly (especially **Soft Bridging**), and added new visualizations and encoder-sensitivity studies. Below we describe how your remarks guided these changes.

---

> ### Author Response · Authors · 2025-11-22
> **Response to Reviewer KEPX (Part 2)**
>
> ### 1. Addressing concerns in model robustness for overlapping / hierarchical / latent-unclear data
>
>
>
> > *“The method assumes that input and output spaces can be cleanly partitioned into clusters that align one-to-one. This may not hold in real-world data with overlapping or hierarchical categories… When clusters poorly capture latent structure (e.g., in WIT), predictive performance degrades notably… limits robustness in high-dimensional or weakly separable spaces.”*
>
>
> We agree with this concern, and now explicitly extend Bridged Clustering beyond the strict one-to-one, clean-cluster regime via two algorithmic variants we newly introduce in Section 8.
>
>
> #### (a) Soft Bridging for non-rigid cluster correspondences
>
> To relax the brittle one-to-one mapping assumption, we introduce **Soft Bridging** (Subsection *Soft Bridging for Flexible Mapping*). We learn a probabilistic bridge:
>
> - For every output cluster $j$, we compute $w_{i \to j}$ as the fraction of supervised points whose inputs fall in cluster $i$, with $\sum_j w_{i \to j} = 1$.
>
> - For an input $x$ assigned to cluster $i$, we predict $\hat y(x)=\sum_{j=1}^{C} w_{i\to j}\bar y_j .$
>
> This removes the strict bijection: **one input cluster now may map to a mixture of output clusters**, which is more realistic in **overlapping, hierarchical, or weakly separable regimes**. Empirically, we validate this design through new **adversarial experiments** that deliberately violate the clean-cluster assumption:
>
> 1. **Overlapping clusters via multi-label categories**, where sampled classes share labels by construction;
> 2. **Mis-specified cluster counts**, where we intentionally use incorrect $K$ in clustering.
>
> In both regimes, Soft Bridging consistently reduces MSE relative to rigid bridging. (**See Figure 2,3 in Appendix A**) We start these adversarial experiments on COCO dataset since the dataset comes with a natural definition of labeled categories and “overlapping” categories of multi-labelled classes, but we can run similar experiments across all datasets if needed.
>
> #### (b) Cluster-wise Regression for richer within-cluster structure
>
> We designed another algorithmic variant that provides a particularly useful template for regimes where $Y$ is **conditionally linear in $X$** given latent $Z$, i.e., beyond purely categorical latents. We will dive specifically into Cluster-wise Regression in the next point --
>
> ---
>
>
>
> ### 2. Addressing concern in unrefined centroid-based predictions
>
>
> > *“Predictions are always the centroid of the output cluster, which may be too coarse for tasks requiring fine-grained outputs. The model does not refine predictions with additional supervision beyond the bridge.”*
>
>
> **Thank you for pushing us to motivate a refinement that restores intra-cluster structure beyond rough centroid prediction.** Your critique echos with other reviewers’ concerns, and pushed our design of **Cluster-wise Regression**.
>
> As we recall, centroid-only Bridged Clustering predicts $\hat y(x)=\bar y_{\hat A(\hat c_X(x))}$, so additional supervision beyond learning $\hat A$ cannot refine per-example outputs. To address this, the new **Cluster-wise Regression** variants restores intra-cluster variation by fitting $f_{i\to j}$ locally and interpolating with centroids:  $\hat y(x)=(1-\alpha)\bar y_j+\alpha f_{i\to j}(x),\quad \alpha\in[0,1].$
>
> Thus, **with enough labels within a cluster pair, predictions become fine-grained**, and supervision continues to help through improved estimation of $f_{i\to j}$.
>
>
> We report improved MSE of this variant under challenging cluster conditions. In the same adversarial settings above, **Cluster-wise Regression (Supervised+Centroid refinement)** improves over centroid-only Bridged Clustering by a small but reliable margin (**See Figure 2,3 in Appendix A**). We clarify that centroid-only Bridged Clustering is ideal in approximately conditionally independent regimes $X \perp Y \mid Z$, while cluster-wise regression targets settings with richer within-cluster structure.

---

> ### Author Response · Authors · 2025-11-22
> **Response to Reviewer KEPX (Part 3)**
>
> ### 3. Addressing Interpretability
>
>
>
> > *“Although interpretability is emphasized, no concrete visualization or case study … is provided.”*
>
>
>
> Thank you for highlighting this gap. We have now added two explicit interpretability visualizations:
>
>
>
> In Appendix D, **Figure 9** illustrates an example of learned cluster bridges on **BIOSCAN**, showing image-DNA species clusters and their sparse correspondences; similarly, **Figure 10** illustrates an example of image-caption learning using Bridged Clustering.
>
>
>
> ---
>
>
>
> ### 4. Addressing Sensitivity to encoder choice and self-supervised embeddings
>
>
>
> > *“Since the method heavily relies on representation separability (∆X, ∆Y), how sensitive is performance to pretrained encoder choice? Have the authors experimented with self-supervised embeddings?”*
>
>
>
> That’s a great question, thank you. In response, we added a new encoder-sensitivity experiment on BIOSCAN, testing **six different image encoders**, including self-supervised models:
>
>
>
> 1. **EfficientNet-B0** (supervised ImageNet pretraining)
>
> 2. **ResNet-50** (supervised ImageNet pretraining)
>
> 3. **Vision Transformer ViT-B/16** (supervised ImageNet pretraining)
>
> 4. **DINO ViT-B/16** (self-supervised DINO pretraining)
>
> 5. **ViT-MAE Base** (self-supervised Masked Autoencoder pretraining)
>
> 6. **DINOv2 ViT-Large** (self-supervised DINOv2 pretraining)
>
>
>
> Results are reported in **Appendix C, Figure 7**. This study suggests credibility in our theoretical analysis (Section 5.2): powerful encoders improve clustering and therefore reduce downstream MSE. Notably, some self-supervised encoders (DINO, DINOv2) perform competitively, supporting the practicality of Bridged Clustering in modern representation pipelines. We sincerely thank the reviewer for prompting us to try self-supervised models that ended up improving BC performance.
>
>
> ---
>
>
>
> ### 5. Theoretical lower bound on labeled pairs for reliable bridging
>
>
>
> > *“Is there a theoretical lower bound on the number of pairs required to guarantee reliable bridging under bounded εX, εY?”*
>
>
>
> Thank you for raising this. This is an excellent direction. When $\varepsilon_X$ and $\varepsilon_Y$ are small enough that each supervised pair lands in its correct input and output clusters with high probability, then the cluster affiliation of that pair is more likely correct than incorrect. Under this condition, for any latent cluster $t$, even **one supervised pair from that cluster** could provide a vote for the true bridge. This explains why we often observe that “single-pair per cluster” suffices once clustering quality is high.
>
> A sharp **information-theoretic lower bound** that explicitly characterizes the minimal pairs needed as a function of $(\varepsilon_X,\varepsilon_Y,C)$ would be very valuable. We now flag this as **future work**, studying a more $\varepsilon$-dependent bound that tightly matches the observed regime where single-pair bridging succeeds.
>
>
>
> ---
>
>
>
> Once again, thank you for your careful and constructive review. Your feedbacks were helpful in our development of:
>
>
>
> - The new **Soft Bridging** and **Cluster-wise Regression** variants for robustness and fine-grained prediction (**Section 8; Appendix A, Figures 2,3**),
>
> - The **adversarial COCO** experiments testing overlapping / miscounted regimes (**Appendix B, Figures 4,6**),
>
> - The new **Figure 9 and 10** interpretability case study,
>
> - And the new **six-encoder sensitivity experiment** including self-supervised representations (**Appendix C, Figure 7**).
>
>
> We hope these additions address your concerns and clarify both the strengths and limitations of Bridged Clustering. Thanks to your constructive feedback, and given the substantial improvements and additional results incorporated during the rebuttal, we hope the revised paper now serves as a meaningful contribution that can inspire further work in this area. We sincerely hope that our reviewers could consider increasing the rating, so that this research can reach and benefit the broader learning community. Thank you so much for your thoughtful engagement.

---

### Official Review · Reviewer_q4Wf · 2025-10-31

**Soundness:** 3
**Presentation:** 3
**Contribution:** 2
**Rating:** 4
**Confidence:** 3

**Summary:**

The authors propose BRIDGED CLUSTERING, an algorithm leveraging both input x, output y data and a small paired (x,y) dataset. The algorithm is very simple, both input and output sets are independently clustered, and an input to output cluster mapping is found through majority voting in the paired dataset. The authors provide a mis-clustering analysis with data generated under sub-gaussian mixtures. The effectiveness is validated empirically on 4 datasets.

**Strengths:**

- The paper is well written and easy to follow. The notation is also coherent and intuitive.
- The algorithm is extremely simple yet well motivated, defined and atomized. The algorithm seems to work well empirically.

**Weaknesses:**

### Major
- My main issue is that all this method reduces all intra cluster variation to 0, which is even more pronounced if the output-only data has high variations. For instance in the image captioning task (L337), any intra-cluster variation in images is lost and all images get the same caption. This strongly limits the applicability of such algorithm. Based on the problem formulation and analysis, the algorithm is only applicable to data derived from categorical latents.

### Minor
- Limited empirical validation.
- L083: “leverages output-only data” this is confusing because it also leverages input data.

**Questions:**

- About the terminology: what does “output-only data mean?” In my understanding, this can only be defined is you define a learner that maps inputs to outputs. I think this should be better explained in the introduction. Because given an unnannotated dataset, there is no notion of input/output. L089 reinforces my claim because you state that you can do bidirectional inference.
- All empirical evaluations seem to be done in a very controlled setup. Why did you limit yourselves to 7 groups? Do you think this would work on more complex datasets?
- L063: “once” should be “if” ?

Overall, I think the paper reads well and my rating would be around borderline. Given that I cannot give a rating of 5, I put 4 as a start but am willing to increase my rating based on the reply and the other reviews.

---

> ### Author Response · Authors · 2025-11-22
> **Response to Reviewer q4Wf (Part 1)**
>
> Thank you so much for your thoughtful and inspiring review. Your comments on increasing intra-cluster variation and solidifying empirical validation genuinely helped us improve our work substantially. In response to your feedback, we added a new section named **“Algorithmic Variants and Extended Applicability”** to the paper. Combining insights from all the reviews, we expanded our experiments to explicitly stress Bridged Clustering under overlapping clusters and misspecified numbers of clusters (including large-cluster-count settings). We also designed two algorithmic variants to address Bridged Clustering's applicability concerns, with **Soft Bridging** performing particularly strongly, and **Cluster-wise Regression** showing promising developments. Below we describe how your remarks guided these changes.

---

> ### Author Response · Authors · 2025-11-22
> **Response to Reviewer q4Wf (Part 2)**
>
> ### 1. Addressing concerns in intra-cluster variation and categorical latents
>
> > *“My main issue is that all this method reduces all intra cluster variation to 0… This strongly limits the applicability of such algorithm. Based on the problem formulation and analysis, the algorithm is only applicable to data derived from categorical latents.”*
>
> Indeed, the original centroid-only Bridged Clustering collapses intra-cluster variation and is most naturally aligned with an approximately categorical latent structure. Addressing this, we present **two concrete algorithmic variants** that extend Bridged Clustering beyond this regime, towards general robustness:
>
> #### (a) Cluster-wise Regression to address intra-cluster variation
>
> Because your main concern is that “all images get the same caption” within a cluster, we were motivated to add an explicit **intra-cluster regression** module (see *Cluster-wise Regression For Refined Prediction* in the new Section 8).
>
> After clustering and bridging, instead of directly returning centroid, we fit **cluster-wise linear regressors** and use them to differentiate predictions for different inputs in the same cluster:
>
> - For every possible pair of input cluster $i$ and output cluster $j$, we fit a linear map $f_{i \to j}$ on all the supervised pairs belonging to that cluster pair.
>
> - We then predict $\hat y(x)=(1 - \alpha)\,\bar y_j+\alpha\, f_{i \to j}(x),\quad \alpha \in [0,1],$
>
> where $\alpha$ controls how strongly the regressor refines the centroid.
>
> We instantiate two variants:
>
> - **Supervised Refinement:** $f_{i \to j}$ is fit on the supervised pairs in $(i,j)$.
>
> - **Supervised+Centroid Refinement:** we augment those supervised pairs with a synthetic "supervised" point created by the input-output cluster centroids $(x_{\mathrm{cen}}, y_{\mathrm{cen}})$.
>
> These variants speak directly to your concern: **within a fixed cluster pair $(i,j)$, different inputs $x$ now receive different predictions** via $f_{i \to j}(x)$, rather than sharing a single centroid output. We also clarify that the original model is best aligned with an close to conditionally independent regime, while **Cluster-wise regression** is most useful where **$Y$ is conditionally linear in $X$** given a latent cluster $Z$, provided enough labels in each cluster to fit these regressors.
>
> We briefly state in the paper that, in our new challenging experiments (overlapping and miscounted clusters), the **Supervised+Centroid Regression** variant indeed improves MSE loss relative to the basic formulation (Appendix A, Figure 2). We start these adversarial experiments on COCO dataset since the dataset comes with a natural definition of labeled categories and “overlapping” categories of multi-labelled classes, but we can run similar experiments across all datasets if needed.
>
> #### (b) Soft Bridging for non-rigid cluster correspondences
>
> We also relax the **rigid one-to-one mapping** between input and output clusters (Section 8: *Soft Bridging for Flexible Mapping*), and introduce **Soft Bridging**.
>
> Instead of assigning each input cluster $i$ to a single output cluster $j$, we learn **probabilistic bridges**:
>
> - For every output cluster $j$, we compute $w_{i \to j}$ as the fraction of supervised points whose inputs fall in cluster $i$, with $\sum_j w_{i \to j} = 1$.
>
> - For an input $x$ assigned to cluster $i$, our model’s prediction becomes $$\hat y(x)=\sum_{j=1}^{C} w_{i \to j}\,\bar y_j.$$
>
> This variant removes the strict 1-1 correspondence, and can handle **overlapping or miscounted clusters** and **poorly separated latents** in data without clear categorical clustering, where a hard permutation is brittle. This addresses your concern over the reliance on clear categorical latents.
>
> **Cluster-wise Regression** and **Soft Bridging** transform the original centroid-only Bridged Clustering algorithm into a robust **family of methods**. We want to thank you for these great suggestions, especially as **Cluster-wise Regression** was directly motivated by your critique of intra-cluster collapse (echoing insights from other reviews), and supported by your observation that our initial formulation fit best with categorical latents.

---

> ### Author Response · Authors · 2025-11-22
> **Response to Reviewer q4Wf (Part 3)**
>
> ### 2. Addressing concerns in limited validation and controlled setups
>
> > *“All empirical evaluations seem to be done in a very controlled setup. Why did you limit yourselves to 7 groups? Do you think this would work on more complex datasets?”*
>
> Your comment aligns with other reviewers' concerns, prompting us both to reexamine the breadth of our original experiment and to add **explicitly adversarial experiments** that increase structural complexity in a controlled way.
>
> #### (a) The original experimental grid
>
> In our original experiment, for each of the four multimodal datasets (BIOSCAN-5M, WIT, Flickr30k, COCO), we run a grid over both the **number of groups** and the **amount of supervision**: Number of Groups $C \in \{3,4,5,6,7\}$, and Supervised Samples per Cluster $\in \{1,2,3,4\}$. This yields **20 settings per dataset**, and for each setting we run **30 randomized trials**, in both directions ($\mathcal{X} \rightarrow \mathcal{Y}$, $\mathcal{Y} \rightarrow \mathcal{X}$) and in both **transductive and inductive** setups.
>
> #### (b) New adversarial experiments in COCO
>
> To go beyond these still somewhat controlled mixtures, we added **two adversarial experiments** that deliberately violate the clean assumptions:
>
> 1. **Overlapping clusters via multi-label categories**: We sampled groups from multi-labeled classes, chosen such that every sampled class shares 1+ label with another sampled class. This explicitly creates overlapping latent groups: images can belong to multiple semantic categories and clusters are not cleanly separable.
>
> 2. **Mis-specified numbers of clusters**: Starting from a known number of sampled categories $K$, we deliberately **mis-specify the number of clusters** used for $k$-clustering in both input and output spaces. This stresses Bridged Clustering when the underlying latent cardinality is not correctly known.
>
> We first ran all our existing methods through the 2 adversarial data conditions, and found that Bridged Clustering does incur more loss in cluster-adversarial conditions, but still maintains overall competitive performance (Appendix B). In the overlapping experiment, Bridged Clustering is competitive against most baselines, second only to one method EOT (See **Figure 4** in Appendix B). Similarly, Bridged Clustering is overall competitive in the mis-specified cluster number experiment, despite the miscount of clusters inevitably undermining its performance (See **Figure 6** in Appendix B).
>
> Encouragingly, when we substituted our existing Bridged Clustering method with the two algorithmic variants we just designed for robustness, we found that they perform well in these cluster-adversarial regimes (See **Figure 2 in Appendix A**). In the Bridged Clustering family of models, we found that the **Soft Bridging** typically achieves the best performance with lowest MSE, while **Cluster-wise Regression (Supervised+Centroid)** wins over the basic Bridged Clustering model by a small but noticeable margin in many cases.
>
> #### \(c\) Overnumbered clusters in BIOSCAN
>
> Besides the above COCO experiments, we also designed **large-cluster-count experiments in BIOSCAN**, which pools in more than the 7 data groups we originally tested (now testing 10, 15, and 20 clusters), since you encouraged us to go beyond 7.
>
> In these BIOSCAN large-cluster-count settings, our existing Bridged Clustering method still works well: MSE remains low and Bridged Clustering stays competitive with strong baselines (Figure 5, Appendix B). Meanwhile, we observe the behavior discussed in our Discussion: as the number of clusters grows, the space of possible cluster mappings increases combinatorially, and our model's performance degrades gradually as cluster numbers grow, consistent with the tradeoff we describe there.
>
> Overall, across COCO and BIOSCAN adversarial settings, we show that: **Cluster-wise Regression** and **Soft Bridging** provide concrete gains precisely in the complex (overlapping and miscounted) regimes you were concerned about.  **Basic Bridged Clustering** remains among the competitive methods even when we mis-cluster, while illustrating the expected sensitivity to imperfect clustering quality and large cluster counts. This discovery conveys an encouraging message -- using the conceptual basis of Bridged Clustering, it's possible to develop a range of algorithms that address different challenging applications, such as when we effectively address cluster-overlaps with Soft Bridges.

---

> ### Author Response · Authors · 2025-11-22
> **Response to Reviewer q4Wf (Part 4)**
>
> ### 3. Terminology and wording
>
> > *“L083: ‘leverages output-only data’ this is confusing because it also leverages input data.”*
>
> > *“What does ‘output-only data’ mean? … given an unannotated dataset, there is no notion of input/output. L089 reinforces my claim because you state that you can do bidirectional inference.”*
>
> > *“L063: ‘once’ should be ‘if’?”*
>
> Thank you for pointing out these issues; we have improved our writing accordingly, clearly explaining what output-only data refers to in our introduction (L43), and clarifying the phrasing in L063 and L083.
>
> ---
>
>
>
> Once again, we are very grateful for your careful and constructive review. Your feedback was essential in pushed our efforts for developing:
>
> - The **Cluster-wise Regression** and **Soft Bridging** variants that restore intra-cluster variation and relax rigid categorical assumptions,
>
> - The **adversarial COCO and BIOSCAN experiments** that probe overlapping and miscounted clusters, along with large-cluster-count cases.
>
> We hope these additions address your concerns and more accurately communicate the scope, flexibility, and limitations of Bridged Clustering.
>
> We believe that Bridged Clustering bears unique domain value. Proposing Bridged Clustering, we envision that this simple formulation has potential for leading into a broader line of research in the joint exploration of unsupervised inputs and outputs. Its conceptual simplicity doesn’t hinder performance (empirically proven), and instead grants it with adaptive potential, as exemplified by the simple but effective variants we designed for more complex applications. We thank you, and the other reviewers, for pushing us to demonstrate this adaptive potential of our work.
>
> Thanks to all of your supportive suggestions, and based on the effortful and promising results this rebuttal has yielded, we hope that this paper has become a valuable resource for inspiring future research. We sincerely hope that our reviewers could consider increasing the rating, and help this work deliver its impact to the learning community. Thank you!

---

> > ### Comment · Reviewer_q4Wf · 2025-11-28
> >
> > I appreciate the effort put in by the authors during the rebuttal period. I am glad that my comments helped improve the manuscript. However, I am somewhat skeptical about the Cluster-wise Regression approach. One aspect I originally liked about Bridged Clustering was its simplicity, whereas the proposed variant introduces an additional regression step that makes the method less straightforward. It is also unclear to me why this two-step approach should outperform a single-step alternative. Figures 2 and 3 in the appendix suggest only limited improvements, especially given the high standard deviations.
> >
> > I thank the authors for the detailed rebuttal. I will maintain my initial rating, though I do not object if the paper is ultimately accepted.

---

> > > ### Author Response · Authors · 2025-12-02
> > > **Response to Comment by Reviewer q4Wf**
> > >
> > > We thank the reviewer again for their thoughtful engagements. Here we add additional clarifications based on the reviewer's newest response.
> > >
> > > Cluster-wise Regression, as a variant to our main formulation, adds a simple regression step before prediction.
> > >
> > > As stated in paper, this variant is especially valuable for extending our simple algorithm beyond low-supervision and conditionally independent settings. In conditionally linear settings, for example, inputs and outputs have direct linear relationships if they appear in bridged clusters (by definition of conditionally linearity). In these settings, when we fit a linear model per cluster (using the available supervised points), we can predict *precisely where* the output should land in our predicted output cluster, rather than simply predicting the cluster centroid. This "supervised refinement" is exactly why the variant outperforms the original and is generally useful, particularly in conditionally linear settings.
> > >
> > > Since we only started experimenting with this variant, we were cautious and set an "influence factor" of 0.25, which means the prediction is only 25% determined by the regression prediction and 75% by the cluster centroid itself (as in our basic formulation). This is why this variant shows some observable improvements, but doesn't appear to be strongly different from the basic method. Furthermore, please note that these experiments are run on the curated very-low-supervised (and not necessarily conditionally linear) COCO dataset, which is not the "natural habitat" for this variant to perform strongly.
> > >
> > > This variant was mainly developed as an example of how Bridged Clustering can be extended to a family of methods addressing different complex data conditions (for example, data with some conditional linearity), and its good performance on existing experiments is a bonus, not central to our main objective.
> > >
> > > Thanks again for all of your thoughtful engagements.

---

### Official Review · Reviewer_Ng8v · 2025-11-01

**Soundness:** 3
**Presentation:** 3
**Contribution:** 3
**Rating:** 6
**Confidence:** 3

**Summary:**

The paper proposes Bridged Clustering (BC), a semi-supervised framework for learning predictors when one has unpaired inputs $X$ and unpaired outputs $Y$ plus a small paired set $S$.

The method: (i) clusters $X$ and $Y$ independently; (ii) learns a sparse cluster-to-cluster bridge from the few paired examples via majority vote; (iii) predicts by assigning a test input to its nearest input cluster and returning the centroid of the linked output cluster (and symmetrically $Y \rightarrow X$).

Experiments on BIOSCAN (bioinformatics) and vision-language datasets (COCO, Flickr30k, WIT) show BC is competitive with SSL, unmatched regression, and transport baselines; it tends to win on BIOSCAN/COCO/Flickr30k and is competitive (but not best) on WIT.

**Strengths:**

1. Focuses on the under-served regime with large unpaired $X$ and $Y$ plus a tiny paired set, using independent clustering and a sparse bridge-distinct from classical SSL and from purely distributional coupling.

2. Model-agnostic encoders and off-the-shelf clustering.

3. Four datasets across modalities; many seeds and settings; BC generally wins on BIOSCAN/COCO/Flickr30k and is competitive on WIT.

**Weaknesses:**

1. Performance hinges on embedding quality and (near-)correct $C$. There is no systematic study of mis-specifying $C$ or robustness to imbalanced/overlapping clusters; theory assumes separation.

2. Majority-vote induces a deterministic mapping $A:[C]\to[C]$. In multi-modal or hierarchical relations, a soft/multi-edge bridge may reduce $\varepsilon_B$.

3. The main metric is embedding-space MSE; this may not fully capture downstream utility.

**Questions:**

1. For captioning/retrieval, could you report Recall@K / median rank and human-readable examples to complement embedding MSE?

2. Section 5.3 claims linear-time scaling once $C \ll n$. Please include wall-clock vs. $n,d$, and $C$ across datasets.

---

> ### Author Response · Authors · 2025-11-22
> **Response to Reviewer Ng8v (Part 1)**
>
> Thank you so much for your thoughtful and inspiring review. Your comments on soft / multi-edge bridges, robustness under misspecification, evaluation metrics, and runtime pushed us to substantially strengthen the paper. Addressing the comments, we present **Section 8, “Algorithmic Variants and Extended Applicability,”**, where we introduce and evaluate a **Soft Bridging** variant directly benefitting by your suggestion, designed adversarial robustness experiments (overlapping and mis-specified clusters, including overnumbered cases), added complementary captioning metrics (BLEU/CHRF), and reported empirical runtime scaling. Below we describe how your remarks guided these changes.

---

> ### Author Response · Authors · 2025-11-22
> **Response to Reviewer Ng8v (Part 2)**
>
> ### 1. Addressing Mapping Formulation
>
>
> > *“Majority-vote induces a deterministic mapping A. In multi-modal or hierarchical relations, a soft/multi-edge bridge may reduce $\epsilon_b$.”*
>
>
>
> We are very grateful for this inspiring comment. Your suggestion of a soft or multi-edge bridge, combined with insights from the other reviews, led us to design the **Soft Bridging** algorithmic variant, which is now one of the strongest versions of Bridged Clustering in our new experiments.
>
>
> In the original method, the bridge is a **hard 1–to-1 mapping** constructed by majority vote over supervised pairs. As you pointed out, this is restrictive in multi-modal, hierarchical, or overlapping relations.
>
> Echoing your observation, we designed **Soft Bridging** (Section *Soft Bridging for Flexible Mapping* in the new Section 8):
>
>
> - We observe each pair of input-output clusters $(i,j)$. For every output cluster $j$, we compute $w_{i \to j}$ as the fraction of supervised points whose inputs fall in cluster $i$, with $\sum_j w_{i \to j} = 1$.
>
> - For an input $x$ assigned to cluster $i$, we predict
>
> $$\hat y(x)=\sum_{j=1}^{C} w_{i\to j}\,\bar y_j .$$
>
>
>
> This implements exactly the **soft / multi-edge bridge** you suggested, removing the rigid bijection and allowing a single input cluster to map to a mixture of output clusters. The method is designed to reduce "mis-mapping" in settings with **multi-modal or hierarchical relations**, and in **overlapping / weakly separated** latent structures.
>
>
> We then evaluated Soft Bridging in **explicitly adversarial experiments**:
>
> 1. **Overlapping clusters** via multi-label categories (latent overlap by construction), and
>
> 2. **Mis-specified cluster counts** (using $K\pm1$, $K\pm2$ instead of the true $K$).
>
>
> In these regimes, **Soft Bridging consistently achieves the lowest MSE among the Bridged Clustering model family** (see **Appendix A, Figures 2–3** for variant comparisons).
>
>
> This variant is largely motivated by your soft-bridge insight and we are thankful for the idea.

---

> ### Author Response · Authors · 2025-11-22
> **Response to Reviewer Ng8v (Part 3)**
>
> ### 2. Addressing concerns in robustness under mis-specified / overlapping / imbalanced clusters
>
>
>
> > *“Performance hinges on embedding quality and (near-)correct C. There is no systematic study of mis-specifying C or robustness to imbalanced/overlapping clusters; theory assumes separation.”*
>
>
>
> We agree that the original experiments did not address robustness to **overlapping** and **mis-specified** cluster structures. Now we add a set of adversarial studies addressing exactly these issues:
>
>
>
> #### (a) Overlapping clusters in COCO
>
>
> To test robustness when separation assumptions are violated, we designed a new COCO experiment. We sampled groups from multi-labeled classes, chosen such that every sampled class shares 1+ label with another sampled class. This creates latent overlap by design: images belong to multiple semantic categories, and clusters cannot cleanly recover disjoint classes.
>
> In this setting, our original Bridged Clustering does lose accuracy but remains competitive—typically second only to EOT among baselines (see **Appendix B, Figure 4**). Crucially, **Soft Bridging** substantially improves robustness here (see **Appendix A, Figure 2, 3**).
>
>
> #### (b) Mis-specified cluster counts in COCO
>
> To study sensitivity to $C$, we next deliberately mis-specify the number of clusters. Let $K$ be the true number of sampled categories -- We run Bridged Clustering and its variants with **$K\pm1$, $K\pm2$** clusters in both input and output spaces. We find:
>
> - Our original Bridged Clustering **remains competitive** across these mis-specified settings, although its performance degrades as miscount increases. It's no longer the "best" method for this setting, but is among the top (**Appendix B, Figure 6**).
>
> - The **Soft Bridging** variant again performs best among Bridged Clustering variants in many of these regimes, mitigating impact of mis-specified $C$ (**Appendix A, Figure 2,3**).
>
> - We designed another algorithmic variant that addresses intra-cluster variability (See Subsection Cluster-wise Regression For Refined Prediction). This variant also improves model performance under both overlapping and miscounting cluster experiments.
>
>
>
> #### \(c\) Overnumbered clusters in BIOSCAN
>
> Finally, to test robustness in substantially overnumbered clusters, we also designed **large-cluster-count experiments in BIOSCAN**, which pools in more than the 7 data groups we originally tested (now testing 10, 15, and 20 clusters). This inflates the size of the cluster mapping space, bringing challenge to the bridging.
>
> The results (see **Appendix B, Figure 5**) show that the **original Bridged Clustering remains competitive and retains low MSE**. Consistent with our Discussion, Bridged Clustering performance becomes less dominantly competitive as $C$ grows. The large-cluster-count BIOSCAN experiments show that our model is robust to moderately large cluster count, but that performance degrades gradually as cluster numbers grow, echoing the tradeoff we describe there.
>
>
> Overall, these adversarial studies directly address your robustness concern:
>
> - **Soft Bridging** and other algorithmic variants provide **concrete performance gains** in overlapping and mis-counted regimes;
>
> - The **basic Bridged Clustering** algorithm remains generally competitive, while performance degrades as expected when the assumptions are strongly violated.

---

> ### Author Response · Authors · 2025-11-22
> **Response to Reviewer Ng8v (Part 4)**
>
> ### 3. Addressing Additional Evaluation and Human-readability
>
>
> > The main metric is embedding-space MSE; this may not fully capture downstream utility.
>
> > For captioning/retrieval, could you report Recall@K / median rank and human-readable examples to complement embedding MSE?
>
>
> Thank you for pushing us to broaden the evaluation perspective. For COCO, Flickr30k, and WIT, we now report **BLEU** scores in **Appendix E** (see **Figure 11-16**). We decode predicted embeddings to captions, then compare to ground-truth captions using BLEU, a metric commonly used in machine translation to evaluate text accuracy.
>
> Across these metrics, Bridged Clustering remain among the leading methods, in line with the trends observed in embedding MSE, especially strong in low supervision (1-2 supervised points per cluster). While our model does not always achieve large absolute gains in BLEU for higher supervision (3-4 supervised points), it is still comparable with the strongest baselines on this text-based retrieval metric.
>
> Our formulation treats output embeddings as **continuous regression targets** and evaluates the **closeness of predicted embeddings to ground-truth embeddings**. Retrieval metrics like Recall@K or median rank are most natural when the task is **ranking a fixed set of candidates** for each query, whereas our experiments focus on learning a predictive mapping in embedding space. Thus we did not add a full Recall@K evaluation pipeline. On the other hand, BLEU on decoded captions gives a complementary, human-readable view of performance. We also illustrate caption examples in Appendix D (Figure 10) to give a more interpretable sense of the bridging process.
>
> We agree that retrieval-style metrics are a natural extension, and we recognize this as an avenue for future empirical work.
>
>
> ---
>
>
>
> ### 4. Addressing Runtime
>
>
>
> > *“Section 5.3 claims linear-time scaling once $C \ll n$. Please include wall-clock vs. $n,d,C$ across datasets.”*
>
>
>
> Thank you for encouraging us to support the runtime claims with empirical evidence. We have now added **wall-clock runtime measurements** in **Appendix F**.
>
> We report wall-clock training time across multiple datasets (COCO, Wiki, Flick30k) as we vary the pool size $n$, and show that **Bridged Clustering exhibits approximate linear runtime growth in $n$ once $C \ll n$**, matching the theoretical scaling in Section 5.3. (We haven't had time to run BIOSCAN yet, but we believe the current experiments suffice in showing approximate linearity.)
>
> We note that our current implementation is **not runtime-optimized**: we did not engineer specialized clustering accelerations or bridge-construction tricks. Choosing different clustering or bridging methods would change runtime substantially. This paper is primarily aimed at the **conceptual algorithmic motivation** of joint exploration in unsupervised inputs and outputs, and we recognize that there is room for empirical and systems-level optimization for downstream engineered applications.
>
> Our claim of linear-time scaling holds asymptotically for substantially large $n>>C$, which is different from the existing empirical setup, where we cannot ignore lower-order terms or constants in runtime.
>
> ---
>
> Once again, thank you for the constructive review. Your suggestion about **soft/multi-edge bridging** was essential in guiding the development of our **Soft Bridging** variant, and your comments on robustness, metrics, and runtime also strengthened the paper. We hope the new Section 8, adversarial experiments (Appendix A-B), metrics (Appendix E), and runtime plots (Appendix F) convincingly address your concerns and make Bridged Clustering a clearer and more robust contribution.
>
> Thanks to your constructive feedback, and given the substantial improvements and additional results incorporated during the rebuttal, we hope the revised paper now serves as a meaningful contribution that can inspire further work in this area. We sincerely hope that our reviewers have increased confidence in accepting this paper, and that this research can reach and benefit the broader learning community. Thank you so much!

---

> > ### Comment · Reviewer_Ng8v · 2025-11-27
> >
> > Thank you for your detailed response. I will maintain my score and still believe the paper should be accepted.

---

> > > ### Author Response · Authors · 2025-12-02
> > > **Response to Comment by Reviewer Ng8v**
> > >
> > > Thank you, your comments were genuinely helpful to our work, and we appreciate your thoughtful engagements.

---

### Official Review · Reviewer_uHM4 · 2025-11-01

**Soundness:** 2
**Presentation:** 2
**Contribution:** 2
**Rating:** 4
**Confidence:** 3

**Summary:**

The article presents Bridged Clustering (BC), a semi-supervised framework for learning mappings between unpaired input–output datasets. The method first clusters each domain independently and then learns a sparse, interpretable “bridge” between clusters using a small set of paired samples. During inference, a new input is assigned to its input cluster and mapped to the centroid of the linked output cluster. The authors provide theoretical analysis (risk bounds under mis-clustering and mis-bridging rates) and evaluate BC across four multimodal datasets (BIOSCAN-5M, WIT, Flickr30k, COCO). The method outperforms or matches strong baselines in most cases while being model-agnostic and computationally efficient.

**Strengths:**

- The paper is clear about its motivation: learning from both input-only and output-only data is an underexplored problem.
- Simple and interpretable design that remains competitive with complex baselines.
- Strong empirical performance on diverse modalities with extremely low supervision.
- Training and inference scale linearly in data size, unlike OT/GW baselines.

**Weaknesses:**

- Method performance is heavily dependent on clustering quality; the paper could discuss robustness or adaptive clustering strategies.

- Theoretical analysis, while rigorous, lacks intuitive explanation or ablation support.

- Limited discussion of failure modes and sensitivity to hyperparameters like the number of clusters.

- Sparse bridge formulation may be too restrictive for many-to-many mappings.

- Baseline tuning is relatively light, might weaken empirical fairness while comparing.
- No sufficient discussion on fixed cluster selection.
- Missing comparison with recent simCLR based methods like SCAN, NNM or so.

**Questions:**

- How sensitive is BC to the choice of cluster count?

- Majority voting seems too simple, could soft or probabilistic bridges (e.g., weighted votes) improve results in overlapping clusters?
- Is it possible to provide runtime comparison in actual time not in complexity?

- How does clustering method choice (k-means vs spectral, DBSCAN, etc.) affect performance?

- Have you tested robustness under noisy embeddings or partially misaligned clusters?

- Could the approach generalize to continuous or hierarchical output spaces?
- What about imbalance dataset?

---

> ### Author Response · Authors · 2025-11-22
> **Response to Reviewer uHM4 (Part 1)**
>
> Thank you so much for your thoughtful and constructive review. Your comments on clustering dependence, robustness, bridging flexibility, runtime evidence, and experimental completeness were all very helpful. The comments were insightful guidances in several substantial upgrades we made to the paper, including a newly added **Section 8: “Algorithmic Variants and Extended Applicability,”**, 3 new adversarial robustness experiments, 2 new algorithmic variants, and expanded empirical diagnostics. Below we describe how your remarks guided these changes.

---

> ### Author Response · Authors · 2025-11-22
> **Response to Reviewer uHM4 (Part 2)**
>
> ### 1. Addressing concerns in dependence on cluster quality
>
>
> > *Method performance is heavily dependent on clustering quality; the paper could discuss robustness or adaptive clustering strategies.*
>
> > *Have you tested robustness under noisy embeddings or partially misaligned clusters?*
>
>
>
> Bridged Clustering is a simple blueprint for joint exploration of unsupervised inputs and outputs. Since we motivate this line of research with simple clustering primitives, we see that clustering quality is a core dependency, and we appreciate you highlighting robustness and adaptivity concerns. We now add both **algorithmic** and **empirical** robustness upgrades:
>
>
> **Algorithmic robustness.** In Section 8, we introduce two variants explicitly designed to reduce brittleness when clusters are imperfect:
>
> 1. **Cluster-wise Regression** restores within-cluster variation and reduces reliance on pure centroid prediction:
>
> $$\hat y(x)=(1-\alpha)\bar y_j+\alpha f_{i\to j}(x),\quad \alpha\in[0,1].$$
>
> This helps when clustering is slightly misaligned (overlapped or miscounted) but still captures coarse semantic structure (See Appendix A, Figure 2, 3).
>
>
> 2. **Soft Bridging** replaces deterministic majority-vote bridges with probabilistic multi-edge bridges:
>
> $$\hat y(x)=\sum_{j=1}^{C} w_{i\to j}\,\bar y_j,\quad \sum_j w_{i\to j}=1,$$
>
> which is explicitly more robust to partial misalignment and overlapping latents (Also See Appendix A, Figure 2, 3).
>
>
>
> **Empirical robustness.** We also added three adversarial experiments that *systematically* create noisy cluster conditions:
>
>
> - **Overlapping-latent COCO experiment** (multi-label categories with constructed overlap) to test weak separability (Appendix B, Figure 4).
>
> - **Mis-specified cluster-number COCO experiment** ($K\pm1$, $K\pm2$) to test bridge reliability under cluster mismatch (Appendix B, Figure 6).
>
> - **Overnumbered BIOSCAN experiment** (10, 15, 20 clusters) to test robustness in large cluster counts (Appendix B, Figure 5).
>
>
>
> Across these settings, the **basic Bridged Clustering remains competitive**, while **Soft Bridging and Cluster-wise Regression provide consistent gains in these misaligned regimes** (Appendix A, Figures 2–3). We now discuss these results as evidence of robustness under imperfect clustering.
>
>
> ---
>
>
> ### 2. Addressing failure modes and sensitivity to cluster count
>
>
>
> > *Limited discussion of failure modes and sensitivity to hyperparameters like the number of clusters.*
>
> > *No sufficient discussion on fixed cluster selection.*
>
> > *How sensitive is BC to the choice of cluster count?*
>
>
> Thank you for emphasizing this gap. As mentioned in the previous point, we now run a **mis-specified cluster-number adversarial study** -- We explicitly run clustering with **incorrect $C$** (both over- and under-counting clusters).
>
> As shown in Appendix B, Figure 6, our original Bridged Clustering **remains somewhat competitive** across these settings, although its performance degrades as miscount increases. It's no longer the "best" method for this setting, but is still among the top (**Appendix B, Figure 6**). Our algorithmic variants bring promising improvements in addressing these conditions, with Soft Bridging obtaining substantial improvement.
>
> This experiment concretely addresses sensitivity to fixed cluster selection and makes the failure modes visible under controlled violations. Algorithmic variants show strong potential for addressing these failure modes.
>
> In addition, results in overnumbered BIOSCAN experiment show that Bridged Clustering is sensitive to large cluster counts --  performance degrades slightly as cluster count grows larger, although still relatively competitive among baselines (Appendix B, Figure 5)
>
> ---
>
> ### 3. Addressing concerns in bridge formulation
>
>
> > *Sparse bridge formulation may be too restrictive for many-to-many mappings. Majority voting seems too simple, could soft or probabilistic bridges improve results in overlapping clusters?*
>
> We strongly agree. This comment, along with similar insights from other reviews, led to us developing a **Soft Bridging** variant (Section 8), which generalizes the bridge from hard 1-to–1 majority voting to a **many-to-many probabilistic bridge** via weights $w_{i\to j}$. Predictions become: $\hat y(x)=\sum_{j=1}^{C} w_{i\to j}\,\bar y_j.$
>
> Empirically, Soft Bridging is most beneficial exactly in the overlapping-latent setting you pointed to, where it consistently reduces MSE relative to original rigid bridging (Appendix A, Figure 2-3).

---

> ### Author Response · Authors · 2025-11-22
> **Response to Reviewer uHM4 (Part 3)**
>
> ### 4. Addressing clustering method choices
>
>
>
> > *How does clustering method choice (k-means vs spectral, DBSCAN, etc.) affect performance?*
>
>
>
> Thank you for raising this. Motivated by your question, we added a direct ablation in **Appendix C, Figure 8**, comparing multiple clustering families in our COCO experiments:
>
>
>
> - **$k$-clustering methods:** K-Means, constrained K-Means, Spectral Clustering, Gaussian Mixture Models, Agglomerative Clustering.
>
> - **Non-$k$ methods:** DBSCAN, Mean Shift, OPTICS.
>
>
>
> We find that **$k$-clustering methods are consistently strongest on our tested datasets**, with K-Means and constrained K-Means performing best overall. In contrast, non-$k$-clustering methods often underperform because they require heavier hyperparameter tuning and may return highly imbalanced or mismatched cluster counts—for example, DBSCAN’s performance is sensitive to neighborhood radius / min-samples and frequently yields too many or too few clusters without dataset-specific tuning. We now include this ablation to clarify that BC is compatible with many clustering primitives, while performance depends on how well the chosen clustering matches the dataset structure.
>
>
>
> ---
>
>
>
> ### 5. Addressing Generalization to continuous or hierarchical output spaces
>
>
>
> > *Could the approach generalize to continuous or hierarchical output spaces?*
>
>
>
> Yes. Our design is representation-level and does not assume discrete outputs; it only requires that the output space admits a meaningful clustering or local structure. The adversarial COCO overlapping experiment (Appendix B, Figure 4) already reflects a step toward hierarchical / multi-label structure, and Soft Bridging explicitly targets this regime.
>
>
>
> ### 6. Addressing Runtime
>
>
> > *Is it possible to provide runtime comparison in actual time not in complexity?*
>
>
> Yes, thank you for requesting this. We added wall-clock runtime plots in **Appendix F**:
>
> We compare the runtime of algorithm across different group numbers. **Figure 10** shows approximate **linear runtime growth in $n$** across multiple datasets once $C\ll n$, matching the claim in Section 5.3. We emphasize that our implementation is **not runtime-optimized** (we did not engineer specialized clustering accelerations or mapping tricks), so constants are not tuned and can affect runtime notably. Alternative clustering/bridging primitives could significantly change runtime. This paper’s focus is mainly on the **theoretical and conceptual basis** of the method; we recognize that systems-level runtime optimization is an important direction for future work.
>
> ---
>
>
>
> ### 7. Addressing Concerns in baseline tuning
>
>
>
> > *Baseline tuning is relatively light, might weaken empirical fairness while comparing.*
>
>
>
> We appreciate this concern. We now clarify that Bridged Clustering itself is a non-tuning method (no task-specific hyperparameter grid beyond shared clustering settings), and all baselines are run under the same supervision / encoder / data splits. Given the sparse-label setting, we prioritized consistent, comparable configurations over heavy per-baseline tuning.
>
>
>
> ---
>
>
>
> ### 8.  Addressing SimCLR-based comparisons and intuitions
>
>
>
>
> > *Missing comparison with recent SimCLR based methods like SCAN, NNM or so.
>
> > Theoretical analysis … lacks intuitive explanation or ablation support.*
>
>
>
> Regarding SimCLR, SCAN , and NNM, we have incorporated discussion and comparisons in the updated related-work. (See L122)
>
>
>
> “Self-supervised contrastive methods like SimCLR involve unlabeled multi-view augmentations, and deep clustering methods such as SCAN and NNM also rely on multi-view consistency. These approaches use multiple augmented views, but operate within a single input modality, while our model jointly targets inputs and outputs.”
>
>
> Bridged Clustering has a fundamentally different learning objective—cross-modal prediction from disjoint $X$-only and $Y$-only corpora—whereas SimCLR-based methods focus on single-modality representation learning and clustering within the input space. As a result, these approaches address different empirical settings and are not directly comparable in experiments.
>
>
> Regarding theoretical analysis, thank you for pointing this out. We strengthen the intuitive explanation of the theory in the revision.

---

> ### Author Response · Authors · 2025-11-22
> **Response to Reviewer uHM4 (Part 4)**
>
> Once again, thank you for the careful and constructive feedback. Your comments, along with insights from all reviewers, have motivated the new robustness variants (**Soft Bridging, Cluster-wise Regression**, Section 8; Appendix A, Figures 2–3), the adversarial misspecification and overlap experiments (Appendix B, Figures 4–6), clustering-method ablation (Appendix C, Figure 8), and wall-clock runtime evidence (Appendix F, Figure 10). All these changes has helped us substantially strengthen our work. We hope these additions address your concerns and better communicate the strengths, limitations, and future directions of Bridged Clustering.
>
> Your constructive feedback has been invaluable, and the improvements and additional experiments we incorporated during the rebuttal have significantly strengthened the paper. We hope the revised version now offers a useful contribution that can motivate further research. We sincerely hope that our reviewers could consider increasing the rating, so that this research can reach and benefit the broader learning community. Thank you sincerely for contributing to our work!

---

### Author Response · Authors · 2025-12-02
**A note to the new AC**

We appreciate your work overseeing this submission. We would like to briefly contextualize the post-rebuttal reviewer discussion.

Following the rebuttal, Reviewer Ng8v stated that our responses fully addressed their concerns and that they believe the paper should be accepted, and they are maintaining their score on that basis. Reviewer q4Wf also acknowledged that our clarifications improved the manuscript and expressed no objection to acceptance, even while maintaining their original score due to lingering skepticism about one variant (Cluster-wise Regression), which is explicitly optional and not central to the main method.

The remaining reviewers did not have the opportunity to post their discussion-phase comments before the security breach and subsequent system freeze, which halted further updates. We believe this is important context for interpreting the partially recorded discussion.

Given that the reviewers who were able to comment indicated that their concerns were resolved and expressed no objection to acceptance, we hope this helps contextualize the current scores relative to the actual discussion.

We appreciate your careful consideration and are happy to provide any additional clarification if helpful.

For convenience, we summarize here the additional experiments, analysis and explanations we added based on the good feedback from our reviewers:

We designed 3 new experiments to test our algorithm’s performance under more complex data conditions: **Overlapping clusters**, **Mis-specified clusters**, **Large-count clusters**. The results directly address the reviewer’s concerns about the algorithm’s generalizability and applicability: Bridged Clustering is still competitive under mis-specified / overlapping / over-numbered cluster conditions.

We designed 2 new algorithmic variants to address reviewer’s inquiry in the algorithm’s flexibility: **Soft Bridging** for non-rigid cluster correspondences, and **Cluster-wise Regression (2 versions)** for richer within-cluster structure. Both variants show strong performance, especially robust in addressing adversarial data in our new robustness experiments. This delivers a convincing message: using the conceptual basis of Bridged Clustering, it’s possible to develop a range of algorithms that address different challenging applications, such as when we effectively address cluster-overlaps with Soft Bridges.

We designed 2 other experiments to prove the algorithm’s robustness under different model choices, per the reviewers’ suggestions. In the **encoder-sensitivity** experiment, we show that Bridged Clustering is robust with different encoders choices. In the **clustering-method** experiment, we show that Bridged Clustering is a true model-agnostic algorithm, robust across different clustering primitives.

We also added more **interpretable metrics** (BLEU scores), **intuitive illustrations and explanations**, and **theory-aligned runtime report**.

All of the above newly added content can be clearly located in our revised paper, concentrated in the new **Section 8** and **Appendices A-F**.

We believe that we have produced a rebuttal that fully addressed all reviewer’s comments and concerns.

Thank you again, the authors.

---

### Meta-Review · Area_Chair_gGtV · 2026-01-04

**Summary:**

The paper proposes a semi-supervised approach, termed Bridged Clustering, for constructing predictive models when inputs and outputs are largely unpaired. The approach separately partitions the input domain and the output domain, and then uses a limited number of matched samples to identify a sparse and interpretable correspondence between the resulting clusters. During prediction, an unseen input is mapped to its corresponding input-side cluster, and the prediction is obtained by selecting the representative element of the associated output-side cluster.

Strengths:

The reviewers praised the motivation, simplicity of the approach, and the good empirical performance. They also noticed that the method works well across a variety of modalities.

Weaknesses:

The reviewers have concerns that performance depends heavily on embedding quality and accurate clustering, with no robustness analysis for mis-specification or overlapping clusters, and that the deterministic majority-vote mapping may be limiting in multi-modal settings. They also pointed out that clustering based mapping also reduces the intra-cluster variation to zero, which does not make a lot of sense for many downstream tasks.

**Reviewer Concerns:**

The authors acknowledge that using clustering based mapping could create issues. To respond to the major concerns, the rebuttal proposed two additional methods, namely, Cluster-wise Regression and Soft Bridging. A new section was added to the manuscript and new experiments were added too. However, a responding reviewer was not convinced.

Although new methods were proposed to counter this concern, the AC feels that the concerns are not fully resolved by comprehensive validation or analysis. The manuscript may use more time and refinement to solidify the newly proposed methods and better position the proposed approaches. The newly proposed methods also may require more thorough experiments to validate.

**Reviewer Scores:**

Reviewer uHM4 (rating 4) has concerns that the method heavily relies on clustering, that the theory lacks ablation, and that there lacks discussion on sensitivity to hyperparameters. There is also a missing comparison to recent SSL methods.  The rebuttal added two new algorithms, ran mis-specified cluster numbers to show robustness/sensitivity, and added more discussions regarding SimCLR etc.  However, the newly proposed algorithms need more comprehensive understanding, and the new mis-specified cluster number is only off by 1 or 2. The reviewer may maintain or increase the score to 6.

Reviewer q4Wf: 4 was concerned about the clustering based mapping. But they were skeptical about the newly proposed Cluster-wise Regression approach in rebuttal and pointed out that the proposed variant introduces an additional regression step that makes the method less straightforward. The reviewer is also not convinced by the performance and wanted to maintain the score.

Reviewer KEPX (rating 4): The reviewer does not like the assumption that the domains can be cleanly clustered. The reviewer does not like centroid based prediction, so the authors proposed cluster-wise regression. Again, the AC feels that these are new sections that are not sufficiently validated. The reviewer may maintain or slightly increase the score.

Reviewer Ng8v (rating 6) also mentioned that there is no systematic study of mis-specifying C or robustness to imbalanced/overlapping clusters, that the theory assumes separation, and that majority voting is deterministic. They also pointed out that there was no downstream metric. The rebuttal added two new algorithms, added robustness experiments with k\pm 1,2 and also BlUE scores are added. The reviewer may maintain the score.

---

### Decision · Program_Chairs · 2026-01-26

Reject